# ON THE COEXISTENCE AND ENSEMBLING OF WATERMARKS

## ABSTRACT

Watermarking, the practice of embedding imperceptible information into media such as images, videos, audio, and text, is essential for intellectual property protection, content provenance and attribution. The growing complexity of digital ecosystems necessitates watermarks for different uses to be embedded in the same media. However, in order to be able to detect and decode all watermarks, they need to coexist well with one another. We perform the first study of coexistence of deep image watermarking methods and, contrary to intuition, we find that various open-source watermarks can coexist with only minor impacts on image quality and decoding robustness. The coexistence of watermarks also opens the avenue for ensembling watermarking methods. We show how ensembling can increase the overall message capacity and enable new trade-offs between capacity, accuracy, robustness and image quality, without needing to retrain the base models.

## 1 INTRODUCTION

Watermarking, the encoding of information into media such as images (Zhu et al., 2018), video (Doerr and Dugelay, 2003; Luo et al., 2023), audio (Hua et al., 2016) or text (Liu et al., 2024) in imperceptible ways, has long been a cornerstone tool for intellectual property protection. While watermarking is not a new technology, it is experiencing a resurgence in interest as a result of the recent growth of AI-generated content and the increased societal expectations and scrutiny (Longpre et al., 2024). Watermarking has been proposed as a tool for restoring stripped content provenance metadata (Collomosse and Parsons, 2024), and for content attribution, where it can be used to trace what training data influenced a newly generated sample (Asnani et al., 2024). In spite of concerns about stripping and spoofing watermarks (Zhao et al., 2023), standards such as C2PA (C2PA, 2024) implement solutions with visual similarity search, manifest databases, and robust fingerprinting.

Watermarking adoption is increasing, especially if legally required as recently proposed (Ricketts, 2023; Wicks, 2024), making it necessary to consider complex ecosystems of watermark providers. However, in a world with many watermarking algorithms, a user —or more likely, their web browser— would not know which detector to use. As such, a sign-posting *super watermark* can be added alongside the original watermark with the goal of indicating how the original watermark has been encoded and what detector should be used to decode it. Furthermore, different actors need to encode different watermarks in the same media, e.g. the author might want to add content provenance watermark, the distributor could apply an intellectual property tracking watermark, and a generative model developer might need a content attribution watermark. Hence, we will likely see more and more cases of multiple watermarks added to the same media.

Embedding multiple watermarks in one image requires their coexistence without mutual interference, enabling independent and accurate decoding of each watermark's message. Yet, the coexistence of multiple deep learning-based watermarks has not been studied, possibly with the assumption that they overwrite each other. That is why we conducted a comprehensive analysis demonstrating they can indeed effectively coexist within the same image. While coexistence incurs minor reductions in image quality and decoding robustness, it persists even when controlling for these factors.

The coexistence of watermarks opens up an avenue for building watermarking ensembles. If two watermarks, with capacity $m_1$ and $m_2$ bits, generated with different methods can both be present in a media, then one can effectively encode $m_1 + m_2$ bits in the image. By combining such watermark ensembling with watermark strength clipping and error-correcting codes, we can modify the

capacity-accuracy-robustness-quality trade-offs of existing methods with no need to retrain them. We show how one can improve existing watermarking techniques by ensembling them with others.

In summary, the contributions of this paper are as follows:

i. We evaluate whether image watermarking techniques can coexist when applied to the same image and demonstrate that coexistence happens to a much greater degree than expected.

ii. We demonstrate that some level of coexistence persists even when one controls for the small quality and robustness degradation resulting of the application of the second watermark.

iii. We show how coexistence enables the ensembling of watermarking models and can be used to modify the performance of existing methods without needing any retraining.

Our experiments focus on image watermarks as this is the domain with most mature watermarking but the same principles likely apply also to video, audio, and to some extent, text watermarking.

## 2  PRELIMINARIES

**Image watermarking.**   Image watermarking is the act of encoding a string of bits (*a secret*) by perturbing the pixel values of an image (*cover image*) in a way that is minimally disruptive and is robust to edits. We will consider the setting when the detector does not have access to the cover image (*blind watermarking*). Watermarking requires a trade-off between four competing objectives:

i. **Capacity:** the length of the secret message string (in bits);

ii. **Image quality:** the amount of distortion added to the cover image to embed the secret, often measured in peak signal-to-noise ratio (PSNR), larger values indicating better quality;

iii. **Accuracy:** the fraction of correctly decoded secrets, usually over a test dataset of images;

iv. **Robustness**: the fraction of secrets we can decode correctly when certain transformations or edits have been added to the image after watermarking. There is no commonly agreed on set of transformations, so we consider the augmentations used for training several popular methods, namely RivaGAN, SSL, TrustMark (low, medium, high), see the details in App. D.

Increasing capacity often lowers the image quality, accuracy, and robustness. Similarly, improving accuracy and robustness typically reduces quality and/or capacity (if using error-correction), while enhancing image quality tends to lower accuracy and robustness. These trade-offs make watermarking a multi-objective problem with no single "best" method, thus only Pareto-optimal solutions can be achieved. In the literature, bit accuracy is often measured instead of full secret accuracy. We believe that full secret accuracy is a better indicator of performance, as recovering the entire message is usually necessary. Therefore, we report the fraction of cases with all bits being correct decoded.

**Watermarking methods.**   *Classical watermarking methods* are based on perturbing the representation of an image in some transform domain: discrete cosine transforms (Dct Tang and Aoki, 1997), discrete wavelet transforms (Dwt Wang et al., 1998, or a combination of the two (DwtDct Al-Haj, 2007 and DwtDctSvd Navas et al., 2008). *Deep watermarking methods* leverage neural networks to do the encoding and decoding of images, leading to more robust watermarks with less distortion with methods such as HiDDeN (Zhu et al., 2018), RivaGAN (Zhang et al., 2019), RoSteALS (Bui et al., 2023b). SSL uses a pretrained image encoder and allows selecting a carrier vector, target image quality and message length at inference time (Fernandez et al., 2022). TrustMark (Bui et al., 2023a) comes in versions with higher accuracy (TrustMark B) and higher quality (TrustMark Q) and has a controllable watermark strength, set to 0.95 by default. *Generative watermarking methods* integrate image generation and watermarking to produce watermarked images directly (Wen et al., 2023; Fernandez et al., 2023). However, since these methods do not produce a non-watermarked image, we cannot measure quality degradation and, therefore, we will not consider them in this work.

## 3  WATERMARKING METHODS CAN COEXIST

**Why might we need more than one watermark in the same image?**   Although media manipulation and disinformation are longstanding issues, recent advancements in generative models have intensified concerns. This has led to a push for comprehensive tools to assure media authenticity (Jones, 2023), often relying on watermarking technology (Gowal and Kohli, 2023; David, 2024),

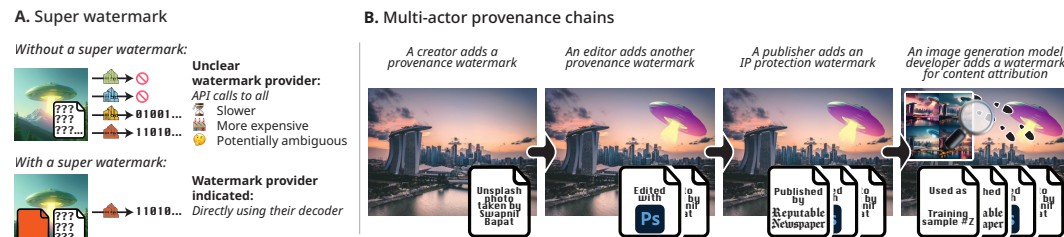

Figure 1: **Two use-cases requiring coexistence of watermarks in the same image. A.** A super watermark informs the user which watermark detector to use for this image. For a super watermark to work, it has to coexist with the main watermark. **B.** As different actors would use different watermarks for content provenance, intellectual property protection and content attribution, these all need to be able to coexist in the same image.

with some proposals to mandate it by law (Ricketts, 2023; Wicks, 2024). With many actors introducing watermarks using diverse techniques, it is challenging for users —and their web browsers— to determine which decoder to use for a given image. For example, C2PA, the most widely adopted content provenance standard, is agnostic to watermarking technology and supports any watermark (C2PA, 2024). Naïvely trying all decoders is impractical, as it requires storing all of them locally or making numerous API calls, which is costly and inefficient. Furthermore, different providers might use the same watermarking method but encode the messages in different ways, leading to multiple watermarks being decoded with only one being the intended one.

We need an efficient way to identify the decoder for a given watermark. One solution is adding a second watermark carrying the provider identifier, similarly to a disk partition table. We call this identification watermark a *super watermark* (see Fig. 1A). With a unified standard for the super watermark, a web browser could support many decoders without evaluating every image against all of them.[1] For a super watermark to be effective, it must *coexist* with the main watermark without hindering its decodability. Another scenario requiring watermark coexistence is the increasingly complex media production and distribution chains, with more and more actors adopting watermarking. For instance, a photo might come with a watermark and editing it in Photoshop might add another to recover its provenance metadata, if stripped. Publishers may add watermarks for copyright detection and developers of generative models might watermark the image to track its contribution to new content and remunerate the copyright holder (Fig. 1B). These varied uses necessitate that watermarks can be applied and decoded independently, i.e., watermarks that can coexist.

These two settings, super watermarks for selecting the correct decoder and multi-actor content provenance chains, depend on different watermarks coexisting in the same image. Yet, to our knowledge, watermark coexistence has not been studied before. While coexistence has been proposed for classical frequency-based methods (Sheppard et al., 2001; Wong et al., 2003; Zear et al., 2018), these techniques do not apply to modern deep-learning-based approaches.

**Watermarks can coexist in the same image.** Considering that watermark coexistence is critical for multi-actor provenance chains and super watermarking, we study to what extent existing open-source techniques can coexist. In the simplest setting, we can apply two watermarks sequentially and measure the accuracy (fraction of correctly decoded secrets) for both methods. We expect the second watermark to be detectable, as the presence of the first should not affect the addition of the second. However, we anticipate that the second watermark would overwrite the first, resulting in 0% accuracy for the first watermark. Similarly, we expect that applying the same watermarking method twice with different secrets would yield low accuracy for the first secret. In Table 1, we present results for 8 watermarking methods, applying each pair in both possible orders. We report the accuracy of the first method, followed by the second. The accuracy when a method is applied alone is shown in grey. When we apply SSL after itself, we use two different carrier vectors.

Our first observation is that no watermarking method can coexist with itself: decoding the first secret yields 0% accuracy across the diagonal in Table 1. In other words, watermarking methods overwrite

---

[1]Different methods tend to have different watermark residuals (see App. E), hence one might consider training a classifier to map images to watermarking methods. Unfortunately, this is impractical because *i)* the classifier would need retraining and redistribution for each new method, and *ii)* different providers might use the same technology but encode messages differently.

Table 1: **Watermarks from different methods can coexist.** We first apply the watermark corresponding to the row, then the one to the column. A new random secret is sampled for every watermark. We report the fraction of secrets with all bits correctly decoded. We show the accuracy of the first method alone (in brackets), followed by the first method when the second is applied, followed by the same for the second method. Surprisingly, for a number of pairs, the application of the second watermark does not overwrite the first watermark. These are the cases where the first two numbers are close, indicating that the accuracy of the first watermark is unaffected when the second is applied. The results are averaged over 1020 samples from *Anon. dataset*.

| | DwtDct | DwtDctSvd | HiDDeN | RivaGAN | RoSteALS | SSL (42dB,30bits) | TrustMark B | TrustMark Q |
|---|---|---|---|---|---|---|---|---|
| DwtDct | (51%)0% /(51%)54% | (51%)59% / (85%)85% | (51%)34% /(15%)15% | (51%)53% /(76%)76% | (51%)50% /(72%)72% | (51%)53% /(94%)93% | (51%)52% /(96%)96% | (51%)53% /(94%)95% |
| DwtDctSvd | (85%)78% /(51%)50% | (85%)0% / (85%)87% | (85%)67% /(15%)16% | (85%)43% /(76%)77% | (85%)5% /(72%)72% | (85%)82% /(94%)97% | (85%)83% /(96%)96% | (85%)82% /(94%)94% |
| HiDDeN | (15%)13% /(51%)55% | (15%)14% / (85%)87% | (15%)0% / (15%)0% | (15%)10% /(76%)76% | (15%)3% /(72%)72% | (15%)15% /(94%)98% | (15%)14% /(96%)96% | (15%)12% /(94%)95% |
| RivaGAN | (76%)75% /(51%)54% | (76%)74% / (85%)93% | (76%)70% /(15%)14% | (76%)0% /(76%)14% | (76%)38% /(72%)70% | (76%)70% /(94%)98% | (76%)71% /(96%)96% | (76%)72% /(94%)94% |
| RoSteALS | (72%)71% /(51%)53% | (72%)65% /(85%)100% | (72%)66% /(15%)14% | (72%)57% /(76%)81% | (72%)0% /(72%)0% | (72%)67% /(94%)99% | (72%)49% /(96%)96% | (72%)51% /(94%)96% |
| SSL (42dB,30bits) | (94%)58% /(51%)53% | (94%)65% /(85%)86% | (94%)36% /(15%)16% | (94%)73% /(76%)79% | (94%)21% /(72%)71% | (94%)0% /(94%)98% | (94%)82% /(96%)95% | (94%)81% /(94%)95% |
| TrustMark B | (96%)96% /(51%)52% | (96%)93% /(85%)95% | (96%)95% /(15%)15% | (96%)94% /(76%)79% | (96%)39% /(72%)66% | (96%)94% /(94%)98% | (96%)0% /(96%)91% | (96%)0% /(94%)94% |
| TrustMark Q | (94%)94% /(51%)52% | (94%)89% /(85%)95% | (94%)90% /(15%)14% | (94%)88% /(76%)77% | (94%)18% /(72%)67% | (94%)86% /(94%)98% | (94%)0% /(96%)95% | (94%)0% /(94%)90% |

their previous message. That holds true also for different but similar methods like TrustMark Q and TrustMark B. Recovering the first secret is unlikely because even if both secrets were theoretically preserved, decoders can output only a single secret. SSL is an exception since both encoder and decoder are conditioned on the carrier vector, yet it too cannot coexist with itself.

Our second observation is the high accuracy for the second method across the board. We see at most a few percentage points drop in accuracy for the second method when two methods are applied. In some cases, we even notice a slight improvement in the accuracy of the second method compared to its solo application, possibly because these methods find it easier to encode noisier images (see columns for DwtDct, DwtDctSvd, RivaGAN, and SSL). Therefore, the presence of a watermark does not hinder adding a new one. The only exceptions are HiDDeN, RivaGAN, and RoSteALS, whose second watermark shows significantly lower accuracy when the same method is applied twice.

Surprisingly, and contrary to our expectations that watermarks overwrite each other, *often the secret of the first watermark can also be decoded with high accuracy!* Since this occurs in most cases, it is more interesting to focus on where coexistence fails. As mentioned earlier, when applying the same watermarking method twice, or when watermarks from the same family are used (e.g., TrustMark Q and TrustMark B or SSL with different carrier vectors), the first watermark's secret seems to be overwritten. RoSteALS overwrites the previous watermark, reducing its accuracy by up to 76% (after TrustMark Q). SSL loses significant accuracy when it is the first method, for example, drop by 58% when applied before HiDDeN. In other cases, the first watermark can be detected with only a moderate drop in accuracy, with lower numbers corresponding to watermarks with lower baseline accuracy. Interestingly, there are cases where the detection of the *first* watermark is improved by the presence of the second one, such as DwtDct followed by DwtDctSvd, RivaGAN, SSL, TrustMark B or TrustMark Q. Therefore, existing watermarking methods coexist well out of the box.

**Channel coding interpretation of why watermark coexistence is surprising.** Watermarking transmits information (the message) over a medium (the image). Traditionally, communication channels are studied in frequency domain (Goldsmith, 2005). Analogous to multiple watermarking, frequency-division multiple access (FDMA) splits the frequency spectrum among users to prevent interference. However, FDMA requires explicit coordination —with authorities dividing the radio frequency spectrum— limiting users to a fraction of the channel's capacity. Watermark coexistence suggests capacity division might be occurring here as well, even without explicit coordination, akin to radio communicators randomly selecting different frequency bands and voluntarily restricting their bandwidth. As deep watermarking is highly non-linear, we cannot directly apply this frequency intuition. Nevertheless, coexistence might indicate that existing methods fail to fully utilize the image capacity to carry information imperceptibly. If a method fully utilized the "channel", it would overwrite any other watermark, which Table 1 indicates to be not happening. In App. A we do offer a possible geometric explanation as to why watermark coexsistence might naturally occur.

**Watermark coexistence does not immediately imply that existing methods are suboptimal.** One might assume that current watermarking techniques are close to Pareto-optimal. However, watermark coexistence suggests that deep watermarking methods may not fully utilize the available capacity, i.e., they fail to use the whole channel. If messages of lengths $m_1$ and $m_2$ can be encoded and decoded using different methods, then the true watermarking capacity is at least $m_1+m_2$, implying that a method reaching this capacity should exist. While tempting, this view is too simplistic. Recall the four-way trade-off among capacity, quality, accuracy, and robustness from Sec. 2. To

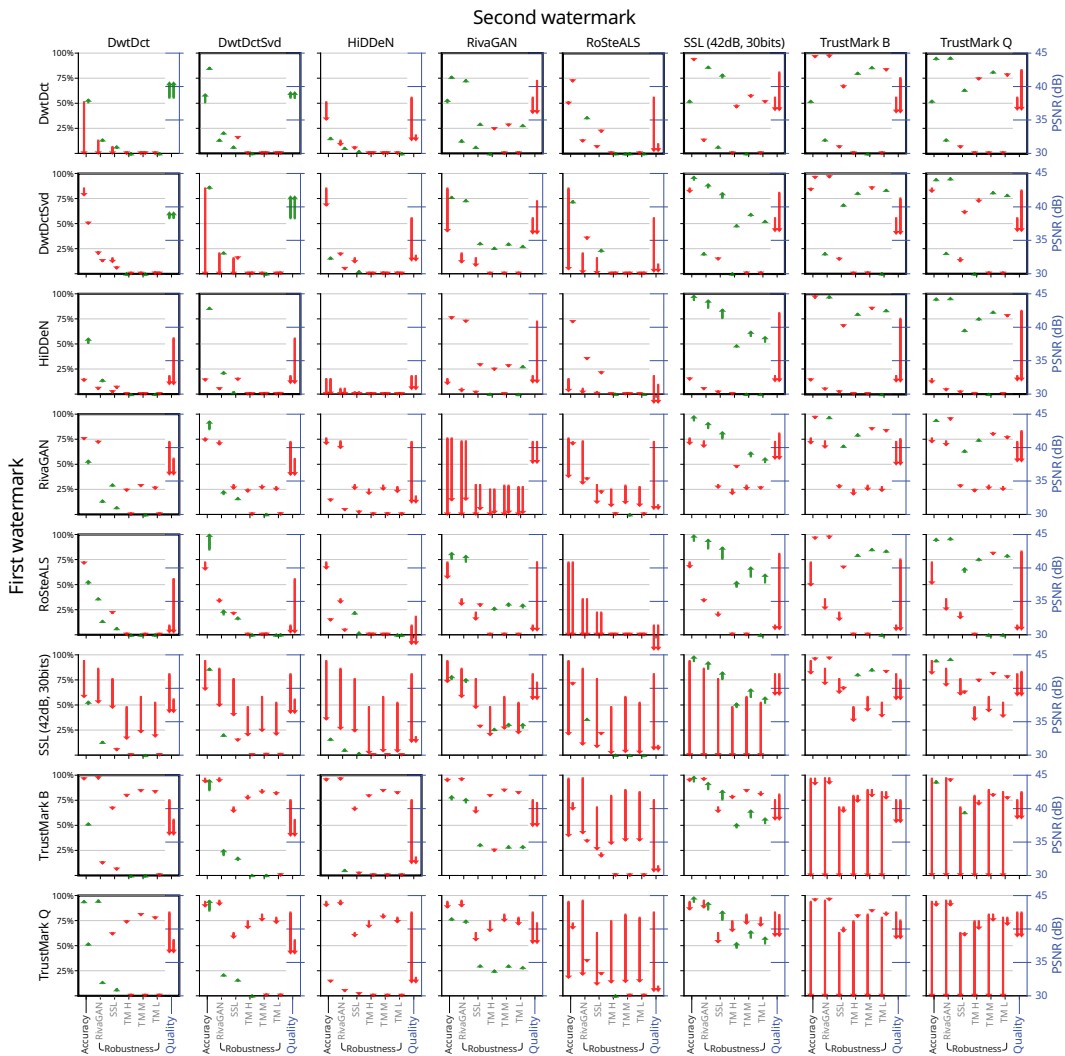

Figure 2: **Watermarks can coexist in the same image.** We show accuracy, robustness and image quality when applying all possible pairs of methods. The left arrow for each metric indicates the change in performance of the first method with and without the second one; vice-versa for the right arrow. In general, two different watermarks can be added to the same image one after the other (in series) and both can be decoded with non-zero accuracy. However, coexistence tends to come at a small cost of accuracy, robustness and image quality. We have highlighted the cases where there is no significant reduction in accuracy and robustness.

argue that the channel is underutilized, the effective increase in capacity that coexistence hints at must not reduce the other three factors. To check whether the capacity gain from ensembling comes at the cost of lower image quality and robustness, in Fig. 2 we show the accuracy, robustness, and quality for both methods across all pairs of watermarking methods. For each metric, we display two arrows: the left for the method applied first and right one for the second. Each arrow starts at the method's standalone value and ends at its value when applied in series. The figure demonstrates that changes in robustness generally track changes in accuracy; when the first method maintains accuracy after the second is applied, it also tends to maintain robustness. There are also cases where the accuracy and robustness of both methods are barely affected by their joint application, for example, DwtDct and DwtDctSvd, HiDDeN followed by SSL, TrustMark B or TrustMark Q, and TrustMark B or TrustMark Q followed by DwtDct (highlighted in Fig. 2). The impact on image quality is more complex. The right-most column in each plot shows that image quality is always worse when two watermarks are applied. Thus, coexistence might involve a trade-off between capacity and quality.

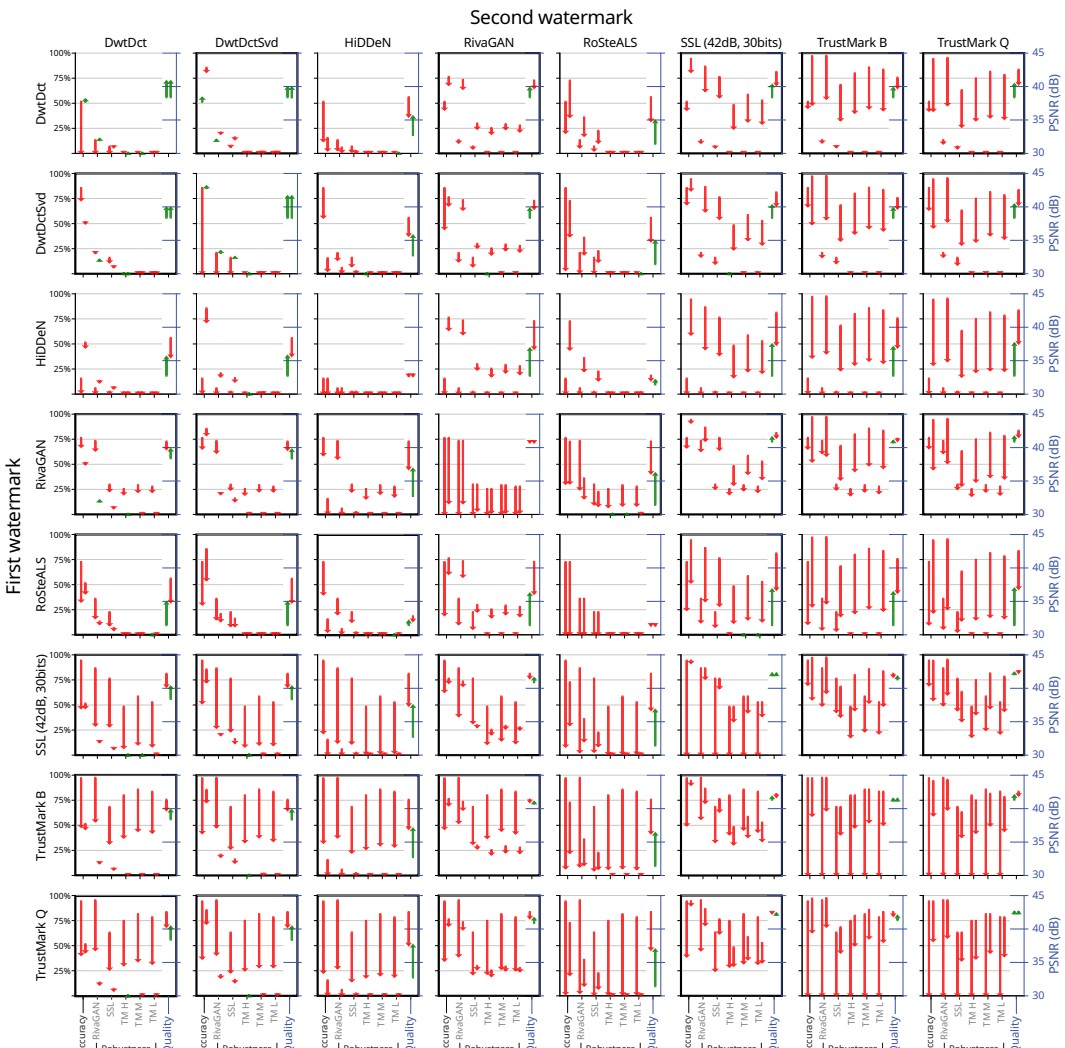

Figure 3: **Controlling for the image quality degradation due to ensembling.** We can reduce the image quality degradation in Fig. 2 due to ensembling by clipping the watermarks to strength 0.5 (see Sec. 4 and Lst. 2 for details. Improving the image quality comes at a further drop in accuracy and robustness but some level of coexistence (i.e., non-zero accuracy for both methods, the endpoints of the leftmost pair of arrows in each cell) persists for most pairs of watermarks. We have highlighted the cases with accuracy ≥25%.

**For many watermarking method pairs, some level of coexistence is preserved even after controlling for image quality.** We can mitigate the lower image quality by reducing the watermark strength so that the final PSNR approximates the mean of the individual methods' PSNRs (details in Lst. 2 with strength=0.5). The results in Fig. 3 show that controlling for the reduced quality results in a non-negligible reduction in accuracy and robustness for most methods though they often are not completely overwritten. As the first set of arrows (corresponding to the detection accuracy) do not go down to zero for most pairs, the accuracy of the first method is not zero (we have highlighted the cases with accuracy ≥25%). *Therefore, the success of coexistence appears to be conditional on the implicit trade-offs between capacity, accuracy, robustness and image quality.*

**The implicit trade-offs underlying watermark coexistence do not explain why it happens in the first place.** At first, the coexistence of watermarks suggests underutilization of channel capacity across methods. However, as shown in Fig. 3, the situation is more nuanced: much of the added capacity comes at the cost of reduced quality and robustness. While these trade-offs indicate that underutilization does not fully explain watermark coexistence, they do not clarify why coexistence

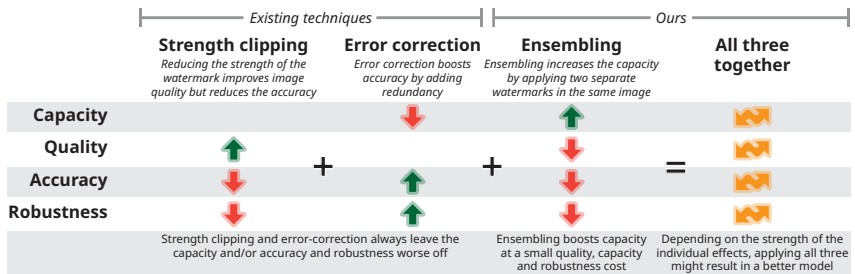

Figure 4: **Comparison of the tools for modifying existing watermarking methods.** Combining strength clipping and ECCs with ensembling, as proposed, can lead to new trade-offs and, possibly, better models.

occurs. Our results suggest that different methods encode information in different, quasi-orthogonal subspaces. It remains an open question why this happens and how to encourage or prevent it.

## 4 ENSEMBLING AS A WATERMARK MODEL MODIFICATION TOOL

Applications of invisible watermarks have varying requirements. An image generator developer might be less concerned about artifacts, since users lack a non-watermarked image for comparison, but may wish to encode extra information like the prompt used. In contrast, content provenance frameworks need high image quality to ensure adoption among creators. Conversely, print media watermarking might need higher robustness to remain detectable after printing and photographing, with lower capacity and quality requirements. Unfortunately, watermarking methods are not easily customizable. Once trained for a specific trade-off between capacity, quality, accuracy and robustness, adjusting this balance typically requires retraining.[2] Consequently, each application might require a slightly different model. Ideally, one would want to modify a watermarking method post-training to adapt it to a new application. We have only seen two tools used for this purpose:

- i. **Strength clip:** One can perform a linear interpolation between the original image and the watermarked image. By bringing the final image closer to the original image, one can improve the image quality, while reducing the detection accuracy and robustness.
- ii. **Error-correcting codes (ECCs):** Error-correcting codes are used for reducing errors in data transmission over unreliable or noisy communication channels by reducing the number of information carrying bits from $n$ to $k$ and introducing $n - k$ redundancy bits in their place (Hamming, 1950). Applying ECCs to the watermarked message reduces the effective capacity of the watermarking method but improves the accuracy and robustness.

While these tools can improve the image quality, accuracy and robustness of existing methods, they *a.* cannot increase the information capacity, and *b.* leave one of the objectives worse off and hence cannot be used to obtain a strictly better watermarking method without retraining.

**Watermark ensembling boosts capacity.** Our observation of watermark coexistence in Sec. 3 offers a potential avenue for a new post-training modification tool: *watermark ensembling*. If the watermarks generated by two different models coexist, then the application of the two methods can be effectively considered to be a new watermarking method having the sum of the capacities of the individual methods. Hence, ensembling directly addresses the first limitation of strength clip and ECCs: it can be used to increase the capacity without retraining (see the third column in Fig. 4). For example, ensembling TrustMark B and SSL (42 dB, 100 bits) gives us 200bit capacity at the cost of lower accuracy and robustness (the capacity doubling can be seen in the first column of Fig. 5A).

**Combining strength clip, ECC and ensembling could, in principle, result in strictly better models.** As seen in Sec. 3, coexistence may come at a small quality, accuracy and robustness cost, thus ensembling alone cannot result in strictly better models. Still, the application of all three tools together could, in principle, outperform either of the two models we start with. Ensembling two watermarking methods coexisting well together increases the capacity to the sum of the individual

---

[2]Exceptions do exist, e.g., SSL allows choosing the secret length and target PSNR at watermarking time.

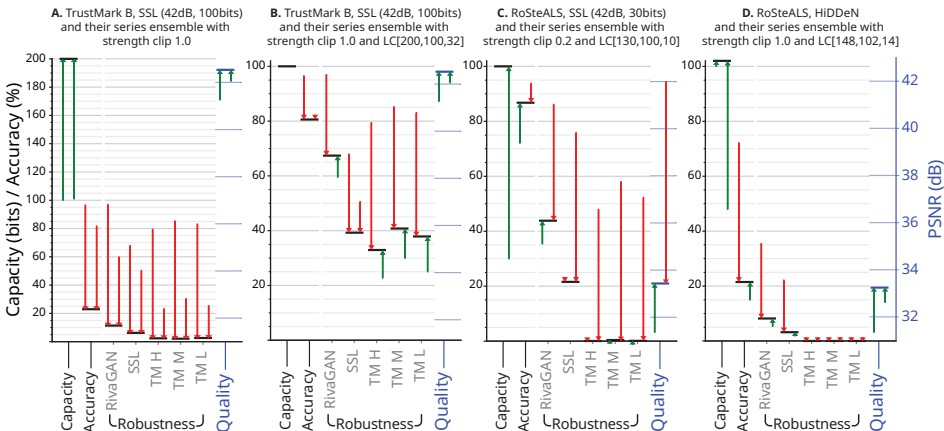

Figure 5: **Ensembling with another method can boost performance.** These plots show two base watermarking methods (the starting points of the left and the right arrows for each pair) and their ensemble with strength clipping and ECC (the horizontal lines/arrow ends). In **A.** the ensemble has significantly larger capacity and image quality, in **B.** ensembling SSL with TrustMark B boosts its robustness and quality, in **C.** ensembling RoSteALS with SSL improves its accuracy and quality, and in **D.** ensembling HiDDeN with RoSteALS boosts its capacity, accuracy, robustness and quality. Numerical values can be found in App. F.

capacities but slightly reduces quality, accuracy and robustness. Clipping the strength of the watermarks will boost image quality and keep the capacity unchanged but may decrease the accuracy and robustness even further. Finally, applying an ECC to the secret can restore the accuracy and robustness, though reducing the effective capacity. Depending on the strength of each of the individual steps, we can produce a various watermarks with no training whatsoever, as illustrated in Fig. 4.

**Experimental setup for testing post-training watermark modifiers.** We consider two ensembling approaches: *series*, where the second watermark is applied to an image already watermarked by the first method (order matters), and *parallel*, where both watermarks are independently applied to the original image, their residuals are averaged, with the result applied to the original image (order doesn't matter). Pseudo code is provided in Lst. 1. To adjust the watermark strength, we apply PSNR clipping where a strength of 0 means the target PSNR is the lower of the individual methods' PSNRs, and a strength of 1 means it is the higher one (pseudo code in Lst. 2). Clipping can also use strengths outside this range. While one could apply the same target PSNR for all images, we opted for an image-wise approach since PSNR values vary greatly between images. We use linear codes (Purser, 1995; Guruswami et al., 2023) with block lengths matching the sum of the capacities of the two base methods. We denote a binary linear code as $LC[n, k, d]$, with $n$ being the block size, $k$ the message size, and $d$ the minimum Hamming distance between codewords. Such a code reduces the effective capacity from $n$ bits to $k$ bits and can correct up to $\lfloor (d-1)/2 \rfloor$ bit flips using the resulting $n - k$ redundancy bits. The codes we use are from the code tables of Grassl (2006).

**Ensembling with another method can boost the watermarking performance.** We offer several examples of improving existing watermarking methods by ensembling them with another method. In Fig. 5B, we show the ensemble of TrustMark B and SSL (42 dB, 100 bits) in series, with clipping with strength 1.0 and $LC[200, 100, 32]$ error-correction bringing the capacity of the ensemble from 100+100=200 bits to 100 bits and being able to correct up to 15 bit flips (details in Lst. 11). While the accuracy and robustness of the ensemble is lower than TrustMark B, it is higher or the same than SSL (except the SSL augmentations, where there is a slight decrease). Furthermore, the quality of the ensembled images is higher than the individual methods, hence ensembling boosted image quality with respect to SSL applied alone while maintaining its capacity, accuracy and robustness.

We also observe cases where a method is improved along all dimensions. For example, in Fig. 5C, we can see that when RoSteALS is ensembled with SSL (42 dB, 30 bits) (series application, 0.2 clipping, $LC[130, 100, 10]$, Lst. 7), we can maintain 100 bit capacity while boosting accuracy by almost 15%, robustness with respect to RivaGAN augmentations with more than 8% and quality with 2dB while maintaining the robustness to SSL augmentations. While SSL alone has higher accuracy, robustness and image quality, its capacity is significantly lower. Similarly, in Fig. 5D

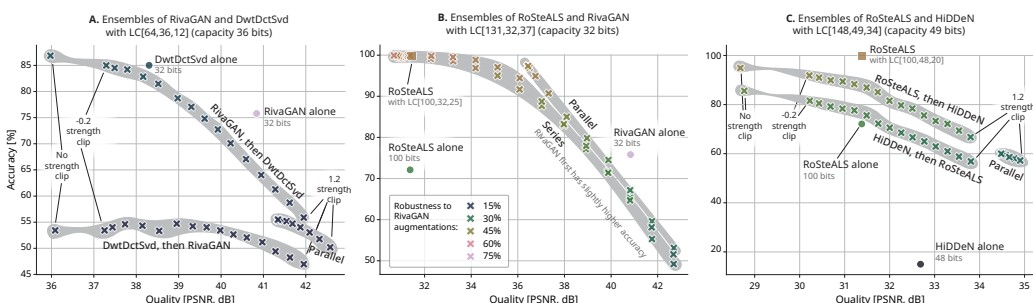

Figure 6: **Ensembling as a tool for fine-tuning watermark trade-offs.** We show how ensembling (together with strength clipping and ECCs) can obtain models with new accuracy–quality trade-offs without further training. In cases when ECC cannot be applied to the baseline models, the ensembles have higher accuracy or better image quality than the base models (**A**). When ECC can be applied to one of the base models, then that can result in higher accuracy than ensembling but ensembling still offers better image quality (**B** and **C**).

we see the capacity of HiDDeN boosted from 48 bits to 102 bits when ensembled with RoSteALS (series, 1.0 clip, LC[148, 102, 14], Lst. 10), with also small improvements in accuracy, robustness, and quality. At first sight RoSteALS might appear to be better than the ensemble due to its higher accuracy and robustness. However, it has noticeably lower image quality and 2 bits less capacity.

**Ensembling as a tool for fine-tuning watermarking trade-offs.** Ensembling not only boosts the watermarking performance but can also do targeted adjustments in order to meet the requirements of new watermarking applications. In Fig. 6 we explore how ensembling achieves various accuracy–quality trade-offs. Each subplot shows ensembles of a pair of watermarking methods: RivaGAN and DwtDctSvd (with LC[64, 36, 12], Lst. 3), RoSteALS and RivaGAN (with LC[131, 32, 37], Lst. 8), and RoSteALS and HiDDeN (with LC[148, 49, 34], Lst. 9). We ensemble them in series and in parallel and apply various levels of strength clipping: from strength -0.2 to strength 1.2. We observe some inherent trade-offs. For example, in Fig. 6A, the base methods have higher accuracy and image quality than the comparable ensembles but 4 bits less capacity. Furthermore, there are ensembles with higher accuracy than the base models (top left corner) and with higher quality than the base models (lower right). In Fig. 6B, while RoSteALS is not on the accuracy–quality Pareto front, it has significantly higher capacity than the ensembles (100 bits vs 32 bits). Finally, in Fig. 6C, we see that the ensembles dominate the base models, RoSteALS and HiDDeN, for all accuracy and quality values. However, RoSteALS has a much higher capacity (100 vs 49 bits). Overall, ensembling enables new trade-offs which would otherwise be accessible only via retraining one of the base models.

**Parallel ensembling tends to result in higher quality while series ensembling seems to result in higher accuracy.** This effect is especially prominent in Fig. 6A and C. As the parallel application multiples the individual watermarks by ½ before combining them, it results in a smaller image perturbation, higher quality and less frequent need for strength clipping. This effect is clear from Fig. 7 which shows that all series-ensembled images have PSNR under 40dB but 5.6% of the parallel-ensembled images have PSNR higher than that. Finally, in Fig. 6A and C series ensembling without strength clip has a drastic reduction of quality, albeit reaching the highest accuracy in both cases.

**In some cases, a strong base method with ECC can perform better than ensembling.** We wish to measure the benefit of ensembling beyond only using ECC and strength clipping. Let's first look at applying ECC alone. We can only apply ECCs when the target capacity is lower than the original model capacity, i.e., when we have spare bits to use for redundancy. Thus we cannot apply ECC to RivaGAN or DwtDctSvd, the base methods in Fig. 6A. In Fig. 6B, we can apply LC[100, 32, 25] (Lst. 4) as RoSteALS has capacity 100 bits but the ensembles just 32 bits. While RoSteALS with LC[100, 32, 25] gets close to 100% accuracy, it maintains the poor image quality of RoSteALS. There are ensembles that have similar accuracy but about 2dB higher quality. Finally, in Fig. 6C, RoSteALS with LC[100, 48, 20] (Lst. 5) does have significantly higher accuracy than the ensembles with similar PSNR. However, there are ensembles that have up to 3.5 dB higher PSNR. Therefore, there are cases where applying ECC to a base models can result in higher accuracy than ensembling with ECC but ECC cannot always be applied (as in Fig. 6A) and cannot improve the image quality (Fig. 6B, C). We also compare whether *both* ECC and strength clipping can outperform ECC, strength clipping

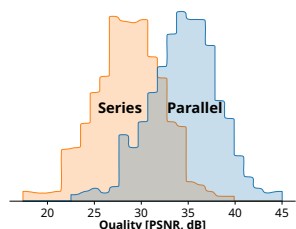

Figure 7: **Ensembling in parallel results in higher image quality than ensembling in series.** The plot shows the distribution of PSNR of images watermarked with ensembles of `RoSteALS` and `HiDDeN` in series and in parallel (without strength clipping) over 1020 images from *Anon. dataset*.

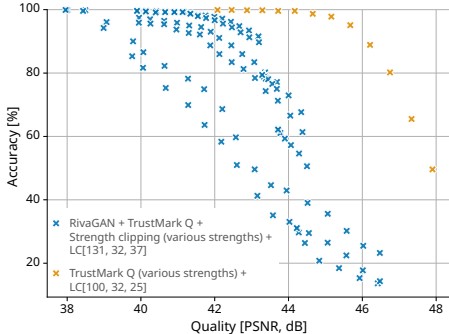

Figure 8: **In some cases, applying ECC alone to a strong base method can perform better than ensembling.** Ensembling `RivaGAN` and `TrustMark Q` (with various levels of strength clipping vs applying `TrustMark Q` at various strengths alone. `TrustMark Q` alone with ECC is better than the ensemble at all quality/accuracy points.

*and* ensembling and find that there are cases where this occurs. For example, in Fig. 8, simply applying $LC[100, 32, 25]$ (Lst. 4) to `TrustMark Q` with various strengths dominates all ensembles of `RivaGAN` and `TrustMark Q`. Therefore, there are also cases when one might be better off simply applying ECC and strength clipping (if possible) to the stronger model than to ensemble them first.

Overall, ensembling is not a one-size-fits-all solution. Ensembling is the only way to increase the capacity of an existing model without retraining. Combined with ECC and strength clipping, ensembling opens new trade-off points but might not result strictly better watermarks. This is likely due to the added capacity failing to compensate for the reduced accuracy and robustness. For example, $LC[200, 100, 32]$ (Lst. 11) uses 100 error-correcting bits but corrects only at most 15 bit flips.

## 5 DISCUSSION AND OPEN QUESTIONS

While we showed that different watermarks can coexist in the same image, we still lack understanding and control of coexistence. Classic frequency analysis offers a good mental model but cannot be directly applied to non-linear deep models. We need a new theory and tools to characterize and evaluate the non-linear channels for encoding information in images, given resolution, quality and robustness constraints. Such a theory would enable coexistence by design: by limiting the channels individual methods use, designing techniques with explicit channel selection allowing multiple coexisting watermarks with the same method, or, conversely, building methods that fail to coexist with any other watermarks thus fully utilising the information-carrying capacity of the image.

Despite ensembling being used frequently for increasing the accuracy of classification and regression tasks (Dong et al., 2020), for boosting robustness (Pang et al., 2019; Petrov et al., 2023), or for achieving computational efficiency for large models (Fedus et al., 2022), we have not seen it applied to watermarking, possibly because it was not obvious that watermarks can coexist. As we showed that watermarks do coexist, we opened an avenue for their ensembling. We observed that ensembling offers nuanced benefits: it boosts the capacity of existing models and open new trade-off points, but sometimes one might be better-off by simply applying ECCs and strength clipping to a single well-performing model as linear codes correct few errors relative to the extra capacity they use. Perhaps, modern ECCs —for instance soft-decision ones which factor in the confidence with which each bit is decoded (Fossorier and Lin, 2002)— could better utilise the extra capacity. Beyond parallel and series ensembling, there could be more advanced adaptive techniques or small learnable mixers that one could use for more performant ensembles. Finally, a further boost to the accuracy and robustness of the ensembles could be possible by fine-tuning their decoders on the jointly watermarked images.

Our analysis was limited in several ways. We used PSNR as a proxy for image quality despite it not tracking human quality perception well; in practice one might want to also consider other quality metrics. Ensembling typically has a computational cost: both the encoding and the decoding steps require inference through both watermarking models. This could be addressed by distilling the two models into a single one. Finally, we focused on images, but watermarking (and hence coexistence and ensembling of watermarks) could be applied to other domains such as video, audio and text.

## REPRODUCIBILITY STATEMENT

To ensure reproducibility of our experiments, we restricted ourselves to watermarking methods with open-source implementations. We used the `invisible-watermark` implementations of DwtDct, DwtDctSvd and RivaGAN and the official implementations of SSL, RoSteALS and TrustMark. For HiDDeN, we used the Stable Signature reimplementation. Furthermore, we provide pseudo-code for our ensembling strategies (in App. B), detailed description of the error-correcting codes we used (in App. C), details about the augmentations for the robustness measures we benchmarked against (in App. D) and comprehensive tables with all the experiments discussed in the paper (in App. F).

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

## A    GEOMETRIC INTUITION AS TO WHY WATERMARKS CAN COEXIST

We would like to offer a geometric intuition as to why watermarks could coexist in the same image without overwriting one another. One explanation is that watermarks are designed to be robust to various augmentations, which induces sparsity into what perturbations can be considered valid watermarks. However, there are many possible sparse patterns that can satisfy the robustness requirements and different watermarking methods might be —either by random selection, or due to particular design decisions— picking different such patterns. As a result, the perturbations used by different methods could be non-overlapping and might be sufficiently different so that they are still identifiable even after composing.

Consider the space of all images $\mathcal{I} = [0,1]^{3 \times h \times w}$. Given an image $x \in \mathcal{I}$, a watermarking method selects a set of images $W(x)$ and maps these images to messages. The capacity of the watermarking method in bits is thus $\log_2 W(x)$. For our illustration we will take $h = w = 1$, and will consider discretized pixel values (as used in practice). Hence, our space of all images $\mathcal{I}$ would look like this:

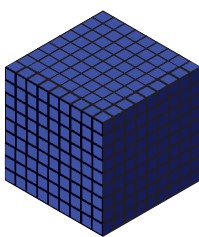

When we watermark an image, we typically want to limit the perturbation $\delta$ applied to an image $x \in \mathcal{I}$ as a quality constraint. For example, a quality constraint formulated as a lower-bound on the PSNR of the watermarked image can be described as limiting $\delta$ to lie in $B_2(x, \epsilon)$, an $\ell_2$ ball centred at $x$ with radius $\epsilon = \sqrt{3wh} \, 10^{\log_{10} R - \text{minPSNR}/10}$, with $R$ being the range of pixel values ($R = 1$ in our case). That means that out of the whole image domain above, we can only use the images near our clean image $x$. The following illustrates these images in blue, with $x$ at the centre of the $3 \times 3 \times 3$ cube:

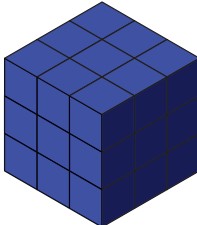

Finally, when watermarking, one typically has robustness constraints as well. That is, we want that the watermarks are identifiable even after they are perturbed a little. Practically, that means that we can only consider subsets of the above blue cube such that all their elements are sufficiently spread out. Formally, the robustness requirement can be formulated as a tolerance relation,[3] that is, a non-transitive equivalence relation $\sim$. For two images $a, b \in \mathcal{I}$ the relation $a \sim b$ holds if there exists an augmentation under the robustness requirement that transforms $a$ into $b$ or vice-versa. For a watermarking method to satisfy such a robustness requirement, the set of images it uses to encode messages cannot contain two images that can be confused under such a transformation, that is:

$$a \nsim b, \qquad \forall a, b \in W(x).$$

However, there can exist many such sets $W(x)$ under the same image quality and robustness constraints.

For the sake of our example, consider that we want our watermarking algorithm to be robust to changing a single pixel value by one discretization step. With this robustness constraint, we can formulate two sets of images that a watermarking scheme might use (with the original image $x$ being marked in red):

---

[3]A *tolerance relation* on a set $A$ is a binary relation $\sim$ that is reflexive ($a \sim a$ for all $a \in A$) and symmetric ($a \sim b \implies b \sim a$ for all $a, b \in A$). See Peters and Wasilewski (2012) for a review on tolerance relations.

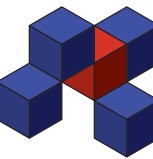 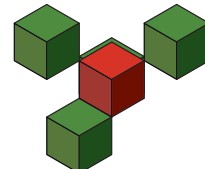

Both sets are maximal: we cannot add an extra element that satisfies both the image quality and the robustness constraints. Moreover, as both sets have size 4, that means that the maximum capacity of watermarking algorithms with these constraints is 2 bits. From the above illustration, one can see how the typical watermarking constraints induce symmetries and sparsity over the sets of images that can be used for watermarking.

Notice also that the two sets are not overlapping and, furthermore, when we add their perturbations, they are still not-overlapping, indicating why coexistence is possible. For example, if we apply the green watermark to marked blue watermarked image, we see that the green and blue images are not overlapping:

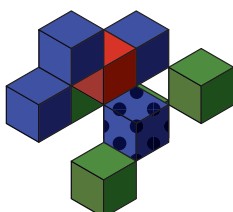

Of course, we also see degradation in the robustness and image quality. In the above illustration, some watermarked images are further away from our original image $x$ (in red) than the $\epsilon$ corresponding to our target PSNR. Furthermore, some watermarked images are next to each other, meaning that the they are no longer robust to augmentations. However, notice that this is "graceful" degradation: only some images have worsened image quality and robustness. This intuition supports the empirical observations we saw in Sec. 3.

It is also important to note that in reality, we operate in higher dimensions with much more complex robustness requirements. Therefore, the sparsity induced in real watermarking applications would likely be much higher than in our toy example. Nevertheless, it remains an open question to perform a formal analysis of watermark coexistence for this general case.

## B  PSEUDO-CODE FOR ENSEMBLING AND STRENGTH CLIPPING

```python
def series_ensemble(
    original: Image,
    wm1: WatermarkingMethod, wm2: WatermarkingMethod, # callable watermarking methods
    m1: List[bool], m2: List[bool] # the binary secrets for the coressponding watermarking methods
) -> Image:

    watermarked1 = wm1(original, m1)
    watermarked2 = wm2(watermarked1, m2)
    return watermarked2

def parallel_ensemble(
    original: Image,
    wm1: WatermarkingMethod, wm2: WatermarkingMethod, # callable watermarking methods
    m1: List[bool], m2: List[bool] # the binary secrets for the coressponding watermarking methods
) -> Image:

    watermarked1 = wm1(original, m1)
    watermarked2 = wm2(original, m2)
    residual1 = watermarked1 - original
    residual2 = watermarked2 - original
    parallel = original + 0.5 * residual1 + 0.5 * residual2
    return clip(parallel, MIN_PIXEL_VALUE, MAX_PIXEL_VALUE)
```

Listing 1: Psudo-code for the series and parallel we use in our ensembling experiments.

```python
def psnr_clip(watermarked: Image, original: Image, target_psnr: float) -> Image:
    diff = watermarked - original
    mse = mean(diff^2)
    scaling_factor = sqrt(10^(2*np.log10(MAX_PIXEL_VALUE-MIN_PIXEL_VALUE) - target_psnr/10.0) / m)
    if scaling_factor >= 1: # if no clipping is necessary
        return watermarked
    return clip(original + scaling_factor * diff, MIN_PIXEL_VALUE, MAX_PIXEL_VALUE)

def clip_to_strength(
    watermarked: Image, original: Image,
    strength: float,
    wm1: WatermarkingMethod, wm2: WatermarkingMethod, # callable watermarking methods
    m1: List[bool], m2: List[bool] # the binary secrets for the coressponding watermarking methods
) -> Image:
    watermarked1 = wm1(original, m1)
    watermarked2 = wm2(original, m2)
    psnr1 = psnr(original, watermarked1)
    psnr2 = psnr(original, watermarked2)

    target_psnr = min(psnr1, psnr2) + strength * (max(psnr1, psnr2) - min(psnr1, psnr2))

    return psnr_clip(watermarked, original, target_psnr)
```

Listing 2: Psudo-code for the strength clipping we use in our ensembling experiments.

# C  CONSTRUCTION OF THE ERROR-CORRECTING CODES

```
1  Construction of a linear code [64,36,12] over GF(2):
2  [1]:  [64, 36, 12] Linear Code over GF(2)
3        Extended BCHCode with parameters 63 11
```

Listing 3: Recipe for constructing a LC[64, 36, 12] binary linear error-correcting code capable of correcting up to 5 bit flips. Recipe taken from CodeTables.de (Grassl, 2006).

```
1  Construction of a linear code [100,32,25] over GF(2):
2  [1]:  [102, 33, 26] Quasicyclic of degree 2 Linear Code over GF(2)
3        QuasiCyclicCode of length 102 with generating polynomials: x^46 + x^44 + x^41 + x^39 + x^35 + x^33 + 1,
           x^46 + x^45 + x^44 + x^43 + x^42 + x^41 + x^40 + x^39 + x^38 + x^37 + x^35 + x^34 + x^33 + x^30 + x
           ^29 + x^26 + x^25 + x^23 + x^22 + x^20 + x^19 + x^17 + x^14 + x^11 + x^10 + x^9 + x^7 + x^6 + x^4
4  [2]:  [101, 32, 26] Linear Code over GF(2)
5        Shortening of [1] at { 102 }
6  [3]:  [100, 32, 25] Linear Code over GF(2)
7        Puncturing of [2] at { 101 }
```

Listing 4: Recipe for constructing a LC[100, 32, 25] binary linear error-correcting code capable of correcting up to 12 bit flips. Recipe taken from CodeTables.de (Grassl, 2006).

```
1  Construction of a linear code [100,48,20] over GF(2):
2  [1]:  [104, 52, 20] Linear Code over GF(2)
3        Extend the QRCode over GF(2)of length 103
4  [2]:  [100, 48, 20] Linear Code over GF(2)
5        Shortening of [1] at { 101 .. 104 }
```

Listing 5: Recipe for constructing a LC[100, 48, 20] binary linear error-correcting code capable of correcting up to 9 bit flips. Recipe taken from CodeTables.de (Grassl, 2006).

```
1  Construction of a linear code [130,30,39] over GF(2):
2  [1]:  [8, 1, 8] Cyclic Linear Code over GF(2)
3        RepetitionCode of length 8
4  [2]:  [127, 29, 43] Cyclic Linear Code over GF(2)
5        CyclicCode of length 127 with Generating Polynomial x^98 + x^91 + x^90 + x^89 + x^86 + x^84 + x^83 + x^82
             + x^80 + x^79 + x^78 + x^77 + x^76 + x^74 + x^72 + x^71 + x^70 + x^65 + x^64 + x^61 + x^59 + x^58
             + x^57 + x^56 + x^51 + x^50 + x^49 + x^45 + x^43 + x^42 + x^40 + x^38 + x^32 + x^31 + x^27 + x^26 +
             x^25 + x^23 + x^18 + x^17 + x^15 + x^13 + x^12 + x^11 + x^9 + x^8 + x^5 + x^4 + x^2 + x + 1
6  [3]:  [127, 30, 35] Linear Code over GF(2)
7        Subcode of dimension 30 between the cyclic codes of length 127 with Generating Polynomials x^91 + x^89 +
             x^87 + x^86 + x^82 + x^80 + x^78 + x^76 + x^75 + x^71 + x^70 + x^67 + x^66 + x^62 + x^60 + x^59 + x
             ^57 + x^56 + x^55 + x^53 + x^52 + x^50 + x^49 + x^48 + x^47 + x^46 + x^45 + x^43 + x^40 + x^37 + x
             ^35 + x^34 + x^33 + x^31 + x^29 + x^28 + x^27 + x^26 + x^25 + x^23 + x^22 + x^20 + x^19 + x^17 + x
             ^16 + x^15 + x^14 + x^13 + x^12 + x^10 + x^9 + x^8 + x^5 + x^4 + 1 and x^98 + x^91 + x^90 + x^89 +
             x^86 + x^84 + x^83 + x^82 + x^80 + x^79 + x^78 + x^77 + x^76 + x^74 + x^72 + x^71 + x^70 + x^65 + x
             ^64 + x^61 + x^59 + x^58 + x^57 + x^56 + x^51 + x^50 + x^49 + x^45 + x^43 + x^42 + x^40 + x^38 + x
             ^32 + x^31 + x^27 + x^26 + x^25 + x^23 + x^18 + x^17 + x^15 + x^13 + x^12 + x^11 + x^9 + x^8 + x^5
             + x^4 + x^2 + x + 1
8  [4]:  [135, 30, 43] Linear Code over GF(2)
9        ConstructionX using [3] [2] and [1]
10 [5]:  [131, 30, 40] Linear Code over GF(2)
11       Puncturing of [4] at { 1, 25, 86, 128 }
12 [6]:  [130, 30, 39] Linear Code over GF(2)
13       Puncturing of [5] at { 131 }
```

Listing 6: Recipe for constructing a LC[130, 30, 39] binary linear error-correcting code capable of correcting up to 19 bit flips. Recipe taken from CodeTables.de (Grassl, 2006).

```
1  Construction of a linear code [130,100,10] over GF(2):
2  [1]:  [8, 7, 2] Cyclic Linear Code over GF(2)
3        Dual of the RepetitionCode of length 8
4  [2]:  [135, 106, 9] Linear Code over GF(2)
5        Let C1 be the BCHCode over GF( 2) of parameters 127 7. Let C2 the SubcodeBetweenCode of dimension 106
             between C1 and the BCHCode with
6  parameters 127 9. Return ConstructionX using C1, C2 and [1]
7  [3]:  [136, 106, 10] Linear Code over GF(2)
8        ExtendCode [2] by 1
9  [4]:  [130, 100, 10] Linear Code over GF(2)
10       Shortening of [3] at { 131 .. 136 }
```

Listing 7: Recipe for constructing a LC[130, 100, 10] binary linear error-correcting code capable of correcting up to 4 bit flips. Recipe taken from CodeTables.de (Grassl, 2006).

```
1  Construction of a linear code [131,32,37] over GF(2):
2  [1]:  [8, 7, 2] Cyclic Linear Code over GF(2)
3        Dual of the RepetitionCode of length 8
4  [2]:  [127, 29, 43] Cyclic Linear Code over GF(2)
5        CyclicCode of length 127 with Generating Polynomial x^98 + x^91 + x^90 + x^89 + x^86 + x^84 + x^83 + x^82
             + x^80 + x^79 + x^78 + x^77 + x^76 + x^74 + x^72 + x^71 + x^70 + x^65 + x^64 + x^61 + x^59 + x^58
             + x^57 + x^56 + x^51 + x^50 + x^49 + x^45 + x^43 + x^42 + x^40 + x^38 + x^32 + x^31 + x^27 + x^26 +
             x^25 + x^23 + x^18 + x^17 + x^15 + x^13 + x^12 + x^11 + x^9 + x^8 + x^5 + x^4 + x^2 + x + 1
6  [3]:  [127, 36, 35] Cyclic Linear Code over GF(2)
7        CyclicCode of length 127 with Generating Polynomial x^91 + x^89 + x^87 + x^86 + x^82 + x^80 + x^78 + x^76
             + x^75 + x^71 + x^70 + x^67 + x^66 + x^62 + x^60 + x^59 + x^57 + x^56 + x^55 + x^53 + x^52 + x^50
             + x^49 + x^48 + x^47 + x^46 + x^45 + x^43 + x^40 + x^37 + x^35 + x^34 + x^33 + x^31 + x^29 + x^28 +
             x^27 + x^26 + x^25 + x^23 + x^22 + x^20 + x^19 + x^17 + x^16 + x^15 + x^14 + x^13 + x^12 + x^10 +
             x^9 + x^8 + x^5 + x^4 + 1
8  [4]:  [135, 36, 37] Linear Code over GF(2)
9        ConstructionX using [3] [2] and [1]
10 [5]:  [131, 32, 37] Linear Code over GF(2)
11       Shortening of [4] at { 132 .. 135 }
```

Listing 8: Recipe for constructing a LC[131, 32, 37] binary linear error-correcting code capable of correcting up to 13 bit flips. Recipe taken from CodeTables.de (Grassl, 2006).

```
1  Construction of a linear code [148,49,34] over GF(2):
2  [1]:  [4, 1, 4] Cyclic Linear Code over GF(2)
3        RepetitionCode of length 4
4  [2]:  [4, 3, 2] Cyclic Linear Code over GF(2)
5        Dual of the RepetitionCode of length 4
6  [3]:  [8, 4, 4] "Reed-Muller Code (r = 1, m = 3)" Linear Code over GF(2)
7        PlotkinSum of [2] and [1]
8  [4]:  [8, 7, 2] Cyclic Linear Code over GF(2)
9        Dual of the RepetitionCode of length 8
10 [5]:  [16, 11, 4] Quasicyclic of degree 4 Linear Code over GF(2)
11       PlotkinSum of [4] and [3]
12 [6]:  [12, 7, 4] Quasicyclic of degree 3 Linear Code over GF(2)
13       Shortening of [5] at { 13 .. 16 }
14 [7]:  [127, 42, 32] Cyclic Linear Code over GF(2)
15       CyclicCode of length 127 with generating polynomial x^85 + x^83 + x^82 + x^77 + x^76 + x^75 + x^74 + x^73
           + x^71 + x^70 + x^69 + x^68 + x^67 + x^65 + x^63 + x^60 + x^57 + x^56 + x^55 + x^54 + x^53 + x^51
           + x^45 + x^44 + x^43 + x^41 + x^40 + x^37 + x^35 + x^31 + x^29 + x^28 + x^27 + x^21 + x^17 + x^15 +
           x^12 + x^9 + x^7 + x^4 + x^3 + x^2 + x + 1
16 [8]:  [127, 42, 32] Cyclic Linear Code over GF(2)
17       CyclicCode of length 127 with generating polynomial x^85 + x^84 + x^83 + x^80 + x^78 + x^76 + x^75 + x^72
           + x^70 + x^68 + x^65 + x^60 + x^58 + x^57 + x^54 + x^53 + x^51 + x^49 + x^47 + x^46 + x^43 + x^39
           + x^38 + x^37 + x^31 + x^30 + x^29 + x^28 + x^24 + x^22 + x^21 + x^19 + x^18 + x^17 + x^15 + x^11 +
           x^6 + x^5 + x + 1
18 [9]:  [127, 49, 28] Cyclic Linear Code over GF(2)
19       CyclicCode of length 127 with generating polynomial x^78 + x^77 + x^76 + x^74 + x^72 + x^67 + x^61 + x^60
           + x^58 + x^57 + x^56 + x^53 + x^52 + x^48 + x^47 + x^45 + x^43 + x^42 + x^35 + x^32 + x^30 + x^26
           + x^25 + x^24 + x^20 + x^19 + x^18 + x^16 + x^13 + x^12 + x^10 + x^5 + x^4 + x^3 + x + 1
20 [10]: [147, 49, 34] Linear Code over GF(2)
21       ConstructionXX using [9] [8] [7] [6] and [4]
22 [11]: [148, 49, 34] Linear Code over GF(2)
23       ExtendCode [10] by 1
```

Listing 9: Recipe for constructing a LC[148, 49, 34] binary linear error-correcting code capable of correcting up to 16 bit flips. Recipe taken from CodeTables.de (Grassl, 2006).

```
1  Construction of a linear code [148,102,14] over GF(2):
2  [1]:  [149, 104, 13] Linear Code over GF(2)
3        Using the construction of Sugiyama with x^6 + a^57*x^5 + a^104*x^4 + a^37*x^3 + a^71*x^2 + a^24*x + a^6
           where a := PrimitiveElement(GF(2,7))
4  [2]:  [150, 104, 14] Linear Code over GF(2)
5        ExtendCode [1] by 1
6  [3]:  [148, 102, 14] Linear Code over GF(2)
7        Shortening of [2] at { 149 .. 150 }
```

Listing 10: Recipe for constructing a LC[148, 102, 14] binary linear error-correcting code capable of correcting up to 6 bit flips. Recipe taken from CodeTables.de (Grassl, 2006).

```
1  Construction of a linear code [200,100,32] over GF(2):
2  [1]:  [199, 100, 31] "Quadratic Residue code" Linear Code over GF(2)
3        QRCode over GF(2)of length 199
4  [2]:  [200, 100, 32] Linear Code over GF(2)
5        ExtendCode [1] by 1
```

Listing 11: Recipe for constructing a LC[200, 100, 32] binary linear error-correcting code capable of correcting up to 15 bit flips. Recipe taken from CodeTables.de (Grassl, 2006).

## D  AUGMENTATIONS USED FOR ROBUSTNESS EVALUATION

Throughout the paper we use five different robustness measures, each corresponding to a set of augmentations. In this section, we outline the specific augmentations that each robustness measure consists of.

**RivaGAN augmentations:**

- Random crop with scale between 0.8 and 1.0 and with probability 0.5.

- Random scale with scale between 0.8 and 1.0 and with probability 0.5.

- Random compress by maintaining between 50% and 100% of the frequencies with probability 0.5.

**SSL augmentations:**

- Random horizontal flip with probability 0.5.

Followed by one transform randomly sampled from the following list:

- Do nothing.

- Random rotation with range (-30, 30).

- Random crop with scale between 0.2 and 1.0 and ratios between $3/4$ and $4/3$.

- Random resize with scale ratio between 0.2 and 1.0.

- Random blur with kernel of maximum size 17.

**Trustmark Low augmentations:**

- Random horizontal flip with probability 0.5.

- Random resized crop with scale between 0.7 and 1.0 and ratios between $3/4$ and $4/3$.

Followed by two transforms randomly sampled from the following list:

- JPEG compression with quality 70 and with probability 0.5.

- Random brightness adjustment to range (0.9, 1.1) with probability 0.5.

- Random contrast adjustment to range (0.9, 1.1) with probability 0.5.

- Random color jiggle with factor (0.05, 0.05, 0.05, 0.01) and with probability 0.5.

- Random grayscale with probability 0.5.

- Random Gaussian blur with kernel size 3, sigma (0.1, 1.0) and with probability 0.5.

- Random Gaussian noise with std 0.02 and with probability 0.5.

- Random hue adjustment with factor (-0.1, 0.1) and with probability 0.5.

- Random motion blur with kernel size (3,5), angle (-25, 25), direction (-0.25, 0.25) and with probability 0.5.

- Random posterize to 5 bits with probability 0.5.

- Random RGB shift with limit for all channels 0.02 and with probability 0.5.

- Random saturation to range (0.9, 1.1) and with probability 0.5.

- Random sharpness to 1.0 and with probability 0.5.

- Random median blur with kernel size 3 and with probability 0.5.

- Random box blur with kernel size 3, border type reflect and with probability 0.5.

**Trustmark Medium augmentations:**

- Random horizontal flip with probability 0.5.
- Random resized crop with scale between 0.7 and 1.0 and ratios between $3/4$ and $4/3$.

Followed by two transforms randomly sampled from the following list:

- JPEG compression with quality 50 and with probability 0.5.
- Random brightness adjustment to range (0.75, 1.25) with probability 0.5.
- Random contrast adjustment to range (0.75, 1.25) with probability 0.5.
- Random color jiggle with factor (0.1, 0.1, 0.1, 0.02) and with probability 0.5.
- Random grayscale with probability 0.5.
- Random Gaussian blur with kernel size 5, sigma (0.1, 1.5) and with probability 0.5.
- Random Gaussian noise with std 0.04 and with probability 0.5.
- Random hue adjustment with factor (-0.2, 0.2) and with probability 0.5.
- Random motion blur with kernel size (3,7), angle (-45, 45), direction (-0.5, 0.5) and with probability 0.5.
- Random posterize to 4 bits with probability 0.5.
- Random RGB shift with limit for all channels 0.05 and with probability 0.5.
- Random saturation to range (0.75, 1.25) and with probability 0.5.
- Random sharpness to 1.5 and with probability 0.5.
- Random median blur with kernel size 3 and with probability 0.5.
- Random box blur with kernel size 5, border type reflect and with probability 0.5.

**Trustmark High augmentations:**

- Random horizontal flip with probability 0.5.
- Random resized crop with scale between 0.7 and 1.0 and ratios between $3/4$ and $4/3$.

Followed by two transforms randomly sampled from the following list:

- JPEG compression with quality 40 and with probability 0.5.
- Random brightness adjustment to range (0.5, 1.5) with probability 0.5.
- Random contrast adjustment to range (0.5, 1.5) with probability 0.5.
- Random color jiggle with factor (0.1, 0.1, 0.1, 0.05) and with probability 0.5.
- Random grayscale with probability 0.5.
- Random Gaussian blur with kernel size 7, sigma (0.1, 2.0) and with probability 0.5.
- Random Gaussian noise with std 0.08 and with probability 0.5.
- Random hue adjustment with factor (-0.5, 0.5) and with probability 0.5.
- Random motion blur with kernel size (3,9), angle (-90, 90), direction (-1, 1) and with probability 0.5.
- Random posterize to 3 bits with probability 0.5.
- Random RGB shift with limit for all channels 0.1 and with probability 0.5.
- Random saturation to range (0.5, 1.5) and with probability 0.5.
- Random sharpness to 2.5 and with probability 0.5.
- Random median blur with kernel size 3 and with probability 0.5.
- Random box blur with kernel size 7, border type reflect and with probability 0.5.

# E    RESIDUALS OF VARIOUS WATERMARKING METHODS

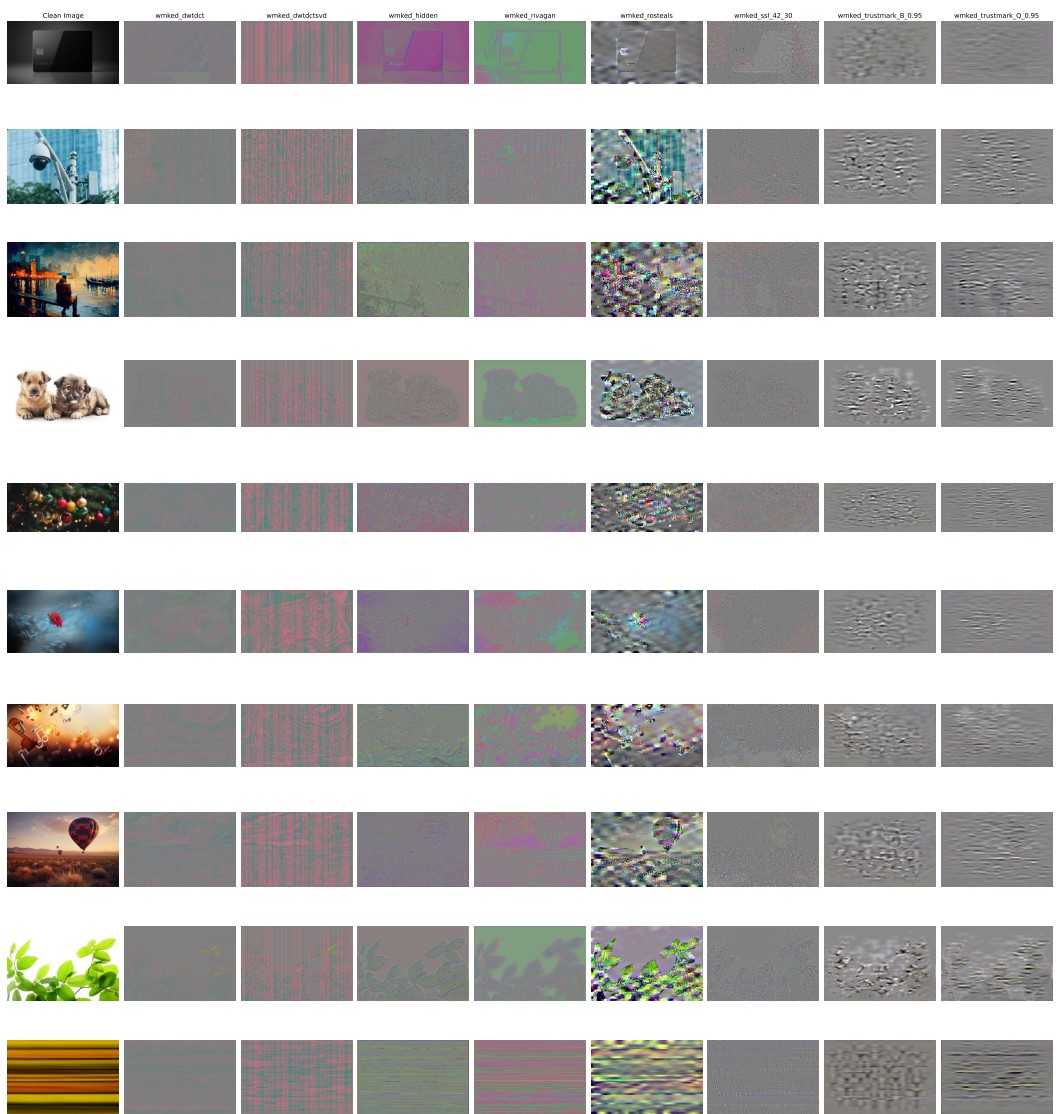

Figure 9: Residuals (difference between the watermarked and the original image) for the 8 watermarking methods studied in the paper. The difference is computed in the RGB space and is scaled 10x. One can observe clear differences in how the different methods perturb the images in order to encode their signal.

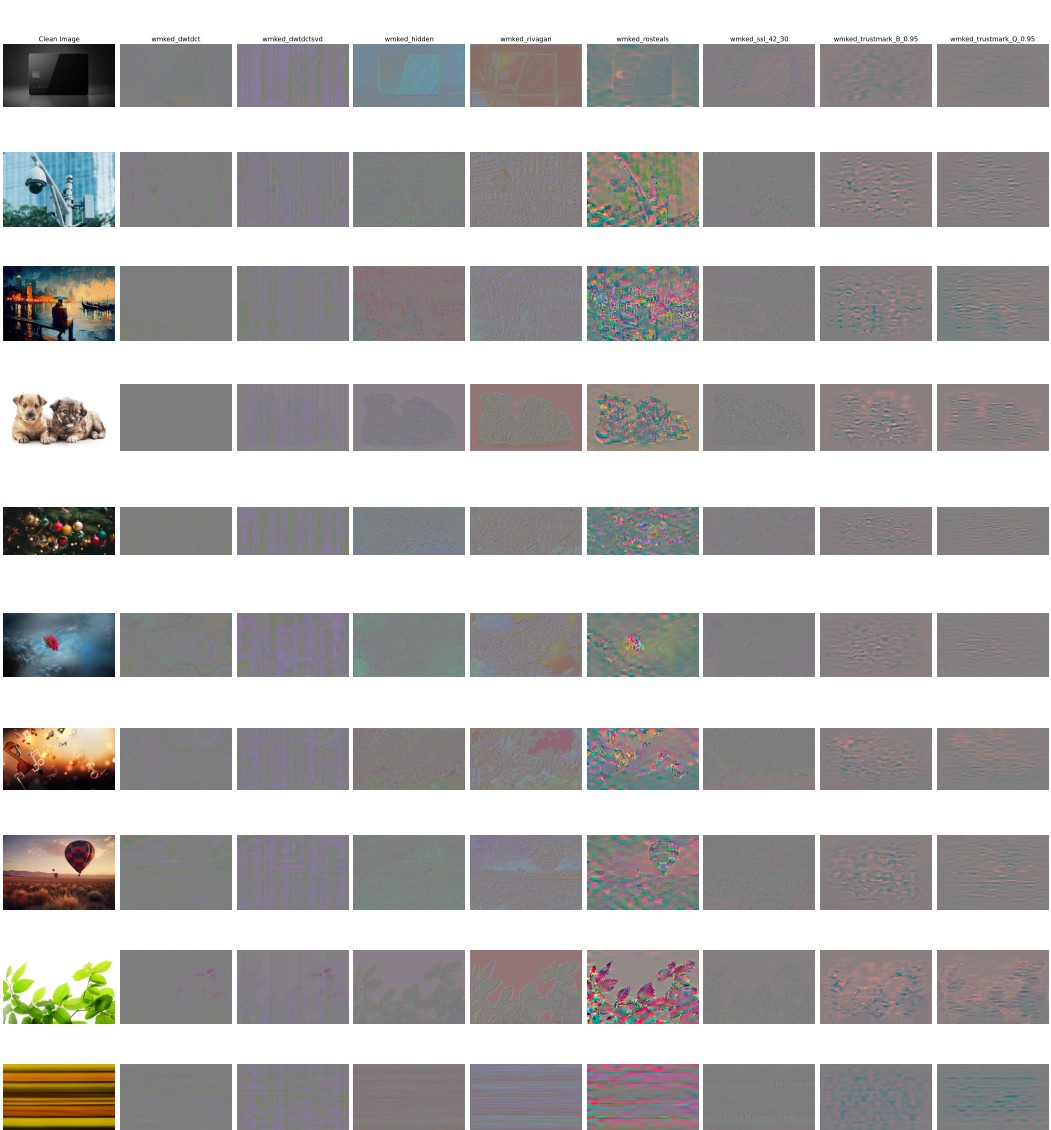

Figure 10: Residuals (difference between the watermarked and the original image) for the 8 watermarking methods studied in the paper. The difference is computed in the YCbCr space and is scaled 10x. One can observe clear differences in how the different methods perturb the images in order to encode their signal.

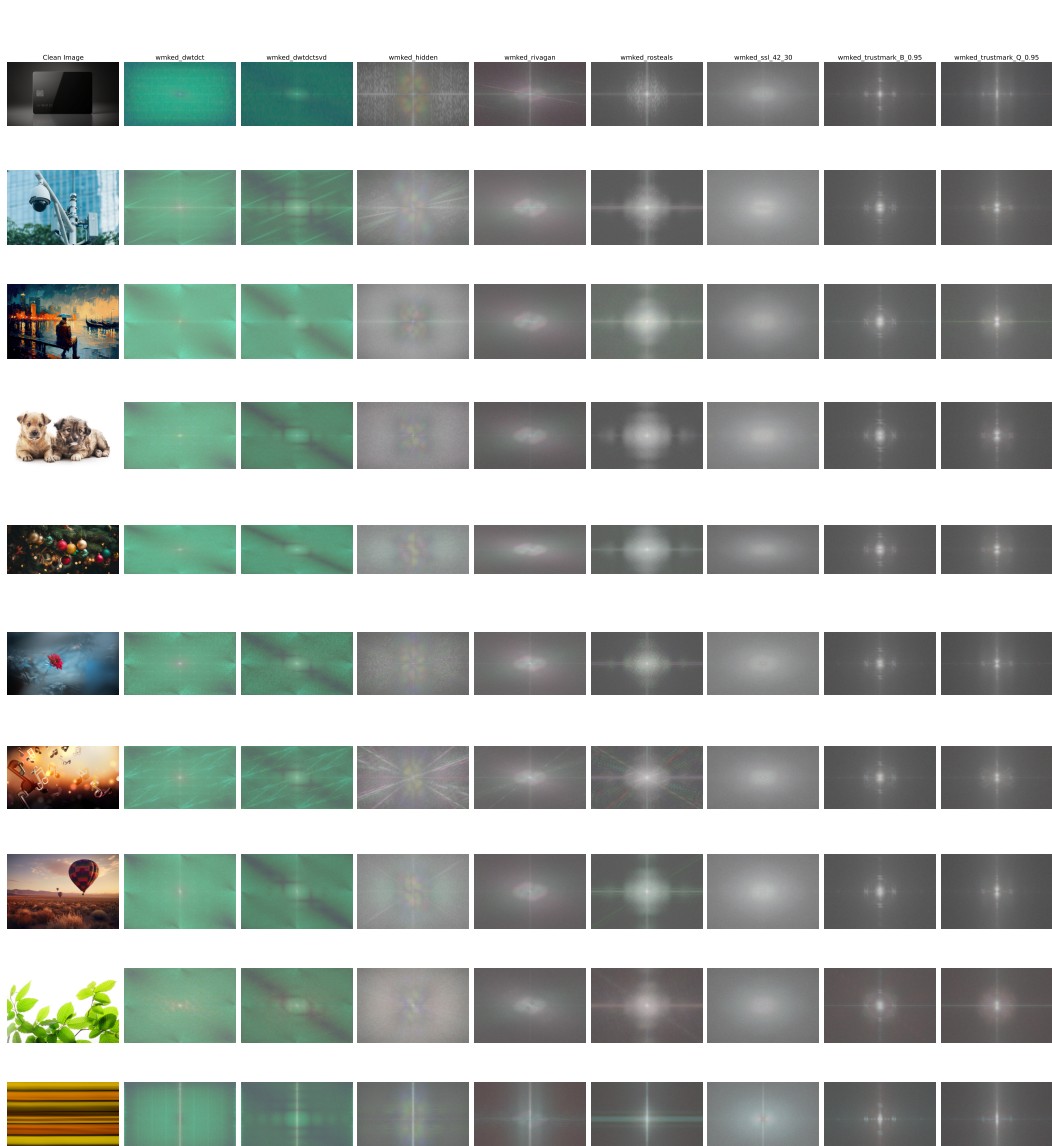

Figure 11: Fourier transform of the residuals (difference between the watermarked and the original image) for the 8 watermarking methods studied in the paper. The difference is computed in the RGB space, the 2D Fourier transform is applied channel-wise, the channels are then scaled with $\sigma(x) = \log(1 + \mathrm{abs}(x))$ and finally normalized to lie between 0 and 255 for producing the images. One can observe clear differences in how the different methods perturb the images in order to encode their signal.

# F  EXTENDED RESULTS

| First method | Second method | ECC | Ensemble | Clip | Capacity (bits) | Accuracy | Mean PSNR [dB] | Std PSNR [dB] | Robust. RivaGAN | Robust. SSL | Robust. TmLow | Robust. TmMed | Robust. TmHigh |
|---|---|---|---|---|---|---|---|---|---|---|---|---|---|
| DwtDct | DwtDct | - | parallel | 0.5 | 64 | 0.0% | 38.7 | 4.1 | 0.0% | 0.0% | 0.0% | 0.0% | 0.0% |
| DwtDct | DwtDct | - | parallel | - | 64 | 0.0% | 38.7 | 4.1 | 0.0% | 0.0% | 0.0% | 0.0% | 0.0% |
| DwtDct | DwtDct | - | series | 0.5 | 64 | 0.0% | 40.7 | 2.3 | 0.0% | 0.0% | 0.0% | 0.0% | 0.0% |
| DwtDct | DwtDct | - | series | - | 64 | 0.0% | 40.5 | 2.2 | 0.0% | 0.0% | 0.0% | 0.0% | 0.0% |
| DwtDct | DwtDctSvd | - | parallel | 0.5 | 64 | 10.0% | 39.4 | 4.5 | 2.9% | 0.9% | 0.0% | 0.0% | 0.0% |
| DwtDct | DwtDctSvd | - | parallel | - | 64 | 10.0% | 39.4 | 4.5 | 2.5% | 1.3% | 0.0% | 0.0% | 0.0% |
| DwtDct | DwtDctSvd | - | series | 0.5 | 64 | 52.9% | 39.8 | 2.5 | 12.6% | 5.8% | 0.0% | 0.0% | 0.0% |
| DwtDct | DwtDctSvd | - | series | - | 64 | 57.2% | 39.1 | 2.0 | 13.3% | 6.4% | 0.0% | 0.0% | 0.0% |
| DwtDct | HiDDeN | - | parallel | 0.5 | 80 | 0.2% | 37.2 | 3.9 | 0.0% | 0.0% | 0.0% | 0.0% | 0.0% |
| DwtDct | HiDDeN | - | parallel | - | 80 | 0.2% | 37.0 | 4.0 | 0.0% | 0.0% | 0.0% | 0.0% | 0.0% |
| DwtDct | HiDDeN | - | series | 0.5 | 80 | 0.6% | 35.5 | 3.5 | 0.0% | 0.0% | 0.0% | 0.0% | 0.0% |
| DwtDct | HiDDeN | - | series | - | 80 | 6.9% | 32.0 | 4.1 | 1.8% | 1.1% | 0.0% | 0.0% | 0.0% |
| DwtDct | - | - |  |  | 32 | 51.0% | 38.3 | 3.8 | 12.7% | 6.4% | 0.0% | 0.1% | 0.0% |
| DwtDct | RivaGAN | - | parallel | 0.5 | 64 | 15.6% | 41.6 | 2.3 | 4.3% | 1.7% | 0.0% | 0.0% | 0.0% |
| DwtDct | RivaGAN | - | parallel | - | 64 | 15.6% | 41.5 | 2.4 | 4.2% | 2.4% | 0.0% | 0.0% | 0.0% |
| DwtDct | RivaGAN | - | series | 0.5 | 64 | 34.5% | 39.7 | 2.1 | 8.4% | 3.3% | 0.0% | 0.0% | 0.0% |
| DwtDct | RivaGAN | - | series | - | 64 | 43.9% | 36.1 | 2.6 | 10.4% | 5.1% | 0.0% | 0.0% | 0.0% |
| DwtDct | RoSteALS | - | parallel | 0.5 | 132 | 2.5% | 36.3 | 3.4 | 0.9% | 0.3% | 0.0% | 0.0% | 0.0% |
| DwtDct | RoSteALS | - | parallel | - | 132 | 2.5% | 36.2 | 3.4 | 0.7% | 0.3% | 0.0% | 0.0% | 0.0% |
| DwtDct | RoSteALS | - | series | 0.5 | 132 | 7.0% | 34.8 | 3.3 | 1.7% | 0.7% | 0.0% | 0.0% | 0.0% |
| DwtDct | RoSteALS | - | series | - | 132 | 35.1% | 30.4 | 3.5 | 8.3% | 4.8% | 0.0% | 0.0% | 0.0% |
| DwtDct | SSL (42dB,30bits) | - | series | 0.5 | 62 | 39.1% | 40.2 | 2.1 | 9.4% | 4.8% | 0.0% | 0.0% | 0.0% |
| DwtDct | SSL (42dB,30bits) | - | series | - | 62 | 49.5% | 36.5 | 2.9 | 11.9% | 6.5% | 0.1% | 0.0% | 0.0% |
| DwtDct | Trustmark B (0.95) | - | parallel | 0.5 | 132 | 2.2% | 41.8 | 2.6 | 0.7% | 0.2% | 0.0% | 0.0% | 0.0% |
| DwtDct | Trustmark B (0.95) | - | parallel | - | 132 | 2.2% | 41.7 | 2.6 | 0.6% | 0.2% | 0.0% | 0.0% | 0.0% |
| DwtDct | Trustmark B (0.95) | - | series | 0.5 | 132 | 20.1% | 39.7 | 2.4 | 4.4% | 2.6% | 0.0% | 0.0% | 0.0% |
| DwtDct | Trustmark B (0.95) | - | series | - | 132 | 49.7% | 36.2 | 2.8 | 12.9% | 5.7% | 0.0% | 0.0% | 0.0% |
| DwtDct | Trustmark Q (0.95) | - | parallel | 0.5 | 132 | 1.8% | 42.1 | 2.7 | 0.6% | 0.2% | 0.0% | 0.0% | 0.0% |
| DwtDct | Trustmark Q (0.95) | - | parallel | - | 132 | 1.8% | 42.1 | 2.8 | 0.5% | 0.3% | 0.0% | 0.0% | 0.0% |
| DwtDct | Trustmark Q (0.95) | - | series | 0.5 | 132 | 15.9% | 40.3 | 2.5 | 3.6% | 1.4% | 0.0% | 0.0% | 0.0% |
| DwtDct | Trustmark Q (0.95) | - | series | - | 132 | 49.5% | 36.6 | 3.0 | 13.2% | 5.6% | 0.0% | 0.0% | 0.0% |
| DwtDctSvd | DwtDct | - | series | 0.5 | 64 | 41.7% | 39.8 | 2.5 | 11.5% | 4.7% | 0.0% | 0.0% | 0.1% |
| DwtDctSvd | DwtDct | - | series | - | 64 | 46.4% | 39.1 | 1.9 | 11.8% | 4.8% | 0.0% | 0.0% | 0.0% |
| DwtDctSvd | DwtDctSvd | - | parallel | 0.5 | 64 | 0.0% | 38.8 | 4.3 | 0.0% | 0.0% | 0.0% | 0.0% | 0.0% |
| DwtDctSvd | DwtDctSvd | - | parallel | - | 64 | 0.0% | 38.8 | 4.3 | 0.0% | 0.0% | 0.0% | 0.0% | 0.0% |
| DwtDctSvd | DwtDctSvd | - | series | 0.5 | 64 | 0.0% | 41.5 | 2.4 | 0.0% | 0.0% | 0.0% | 0.0% | 0.0% |
| DwtDctSvd | DwtDctSvd | - | series | - | 64 | 0.0% | 41.5 | 2.3 | 0.0% | 0.0% | 0.0% | 0.0% | 0.0% |
| DwtDctSvd | HiDDeN | - | parallel | 0.5 | 80 | 0.1% | 37.1 | 3.8 | 0.0% | 0.0% | 0.0% | 0.0% | 0.0% |
| DwtDctSvd | HiDDeN | - | parallel | - | 80 | 0.1% | 37.0 | 4.0 | 0.0% | 0.0% | 0.0% | 0.0% | 0.0% |
| DwtDctSvd | HiDDeN | - | series | 0.5 | 80 | 2.3% | 35.4 | 3.5 | 0.5% | 0.2% | 0.0% | 0.0% | 0.0% |
| DwtDctSvd | HiDDeN | - | series | - | 80 | 11.6% | 31.8 | 3.9 | 2.6% | 1.7% | 0.0% | 0.0% | 0.0% |
| DwtDctSvd | - | - |  |  | 32 | 85.0% | 38.3 | 3.9 | 20.3% | 15.5% | 0.3% | 0.2% | 0.1% |
| DwtDctSvd | RivaGAN | LC[64,36,12] | parallel | -0.2 | 36 | 55.5% | 41.3 | 2.3 | 13.8% | 9.7% | 0.1% | 0.0% | 0.0% |
| DwtDctSvd | RivaGAN | LC[64,36,12] | parallel | -0.1 | 36 | 55.5% | 41.3 | 2.3 | 12.9% | 10.0% | 0.2% | 0.3% | 0.0% |
| DwtDctSvd | RivaGAN | LC[64,36,12] | parallel | 0 | 36 | 55.5% | 41.3 | 2.3 | 13.6% | 9.0% | 0.0% | 0.1% | 0.0% |
| DwtDctSvd | RivaGAN | LC[64,36,12] | parallel | 0.1 | 36 | 55.5% | 41.3 | 2.3 | 14.5% | 9.4% | 0.1% | 0.0% | 0.1% |
| DwtDctSvd | RivaGAN | LC[64,36,12] | parallel | 0.2 | 36 | 55.5% | 41.3 | 2.3 | 14.5% | 10.6% | 0.1% | 0.0% | 0.0% |
| DwtDctSvd | RivaGAN | LC[64,36,12] | parallel | 0.3 | 36 | 55.5% | 41.3 | 2.3 | 13.8% | 10.9% | 0.1% | 0.0% | 0.1% |
| DwtDctSvd | RivaGAN | LC[64,36,12] | parallel | 0.4 | 36 | 55.5% | 41.3 | 2.3 | 13.2% | 9.0% | 0.0% | 0.1% | 0.0% |
| DwtDctSvd | RivaGAN | LC[64,36,12] | parallel | 0.5 | 36 | 55.4% | 41.4 | 2.2 | 14.7% | 9.8% | 0.1% | 0.1% | 0.0% |
| DwtDctSvd | RivaGAN | LC[64,36,12] | parallel | 0.6 | 36 | 55.3% | 41.5 | 2.2 | 13.9% | 9.4% | 0.1% | 0.2% | 0.2% |
| DwtDctSvd | RivaGAN | LC[64,36,12] | parallel | 0.7 | 36 | 55.2% | 41.5 | 2.0 | 13.3% | 8.8% | 0.3% | 0.1% | 0.0% |
| DwtDctSvd | RivaGAN | LC[64,36,12] | parallel | 0.8 | 36 | 54.7% | 41.7 | 1.9 | 14.1% | 10.2% | 0.1% | 0.1% | 0.1% |
| DwtDctSvd | RivaGAN | LC[64,36,12] | parallel | 0.9 | 36 | 54.0% | 41.9 | 1.8 | 12.9% | 9.9% | 0.0% | 0.0% | 0.0% |
| DwtDctSvd | RivaGAN | LC[64,36,12] | parallel | 1 | 36 | 53.0% | 42.1 | 1.8 | 12.3% | 8.9% | 0.2% | 0.1% | 0.1% |
| DwtDctSvd | RivaGAN | LC[64,36,12] | parallel | 1.1 | 36 | 51.8% | 42.3 | 1.7 | 13.0% | 9.2% | 0.0% | 0.0% | 0.1% |
| DwtDctSvd | RivaGAN | LC[64,36,12] | parallel | 1.2 | 36 | 50.2% | 42.6 | 1.7 | 12.5% | 8.9% | 0.0% | 0.0% | 0.0% |
| DwtDctSvd | RivaGAN | LC[64,36,12] | series | -0.2 | 36 | 55.5% | 41.3 | 2.3 | 14.1% | 10.0% | 0.1% | 0.4% | 0.0% |
| DwtDctSvd | RivaGAN | LC[64,36,12] | series | -0.1 | 36 | 53.4% | 37.3 | 3.3 | 13.1% | 10.5% | 0.2% | 0.2% | 0.1% |
| DwtDctSvd | RivaGAN | LC[64,36,12] | series | 0 | 36 | 54.0% | 37.4 | 3.3 | 13.5% | 10.2% | 0.1% | 0.2% | 0.1% |
| DwtDctSvd | RivaGAN | LC[64,36,12] | series | 0.1 | 36 | 54.6% | 37.7 | 3.2 | 12.7% | 9.5% | 0.4% | 0.1% | 0.0% |
| DwtDctSvd | RivaGAN | LC[64,36,12] | series | 0.2 | 36 | 54.3% | 38.1 | 3.0 | 14.7% | 9.7% | 0.1% | 0.1% | 0.2% |
| DwtDctSvd | RivaGAN | LC[64,36,12] | series | 0.3 | 36 | 53.3% | 38.5 | 2.8 | 13.1% | 9.3% | 0.1% | 0.2% | 0.0% |
| DwtDctSvd | RivaGAN | LC[64,36,12] | series | 0.4 | 36 | 54.7% | 39.0 | 2.5 | 14.0% | 10.0% | 0.0% | 0.1% | 0.1% |
| DwtDctSvd | RivaGAN | LC[64,36,12] | series | 0.5 | 36 | 54.2% | 39.3 | 2.3 | 13.5% | 8.8% | 0.2% | 0.0% | 0.0% |
| DwtDctSvd | RivaGAN | LC[64,36,12] | series | 0.6 | 36 | 54.0% | 39.7 | 2.1 | 14.1% | 9.5% | 0.1% | 0.1% | 0.2% |
| DwtDctSvd | RivaGAN | LC[64,36,12] | series | 0.7 | 36 | 53.4% | 40.0 | 2.0 | 12.7% | 9.3% | 0.1% | 0.0% | 0.1% |
| DwtDctSvd | RivaGAN | LC[64,36,12] | series | 0.8 | 36 | 52.6% | 40.3 | 1.8 | 12.5% | 10.4% | 0.1% | 0.1% | 0.0% |
| DwtDctSvd | RivaGAN | LC[64,36,12] | series | 0.9 | 36 | 52.1% | 40.6 | 1.7 | 13.2% | 9.1% | 0.2% | 0.0% | 0.0% |
| DwtDctSvd | RivaGAN | LC[64,36,12] | series | 1 | 36 | 51.2% | 41.0 | 1.8 | 13.1% | 8.4% | 0.0% | 0.0% | 0.2% |
| DwtDctSvd | RivaGAN | LC[64,36,12] | series | 1.1 | 36 | 49.4% | 41.3 | 1.8 | 12.0% | 9.2% | 0.0% | 0.0% | 0.0% |
| DwtDctSvd | RivaGAN | LC[64,36,12] | series | 1.2 | 36 | 48.2% | 41.6 | 1.8 | 12.1% | 8.7% | 0.1% | 0.0% | 0.1% |
| DwtDctSvd | RivaGAN | LC[64,36,12] | series | - | 36 | 47.0% | 42.0 | 1.8 | 12.0% | 8.2% | 0.1% | 0.1% | 0.0% |
| DwtDctSvd | RivaGAN | LC[64,36,12] | series | - | 36 | 53.4% | 36.1 | 2.6 | 14.6% | 10.4% | 0.0% | 0.2% | 0.0% |
| DwtDctSvd | RivaGAN | - | parallel | 0.5 | 64 | 34.5% | 41.4 | 2.2 | 9.3% | 4.9% | 0.0% | 0.0% | 0.0% |
| DwtDctSvd | RivaGAN | - | parallel | - | 64 | 34.5% | 41.3 | 2.3 | 8.8% | 5.0% | 0.0% | 0.0% | 0.0% |
| DwtDctSvd | RivaGAN | - | series | 0.5 | 64 | 32.8% | 39.7 | 2.1 | 8.9% | 4.7% | 0.0% | 0.1% | 0.0% |
| DwtDctSvd | RivaGAN | - | series | - | 64 | 36.4% | 36.1 | 2.6 | 9.8% | 6.9% | 0.0% | 0.0% | 0.0% |
| DwtDctSvd | RoSteALS | - | parallel | 0.5 | 132 | 5.2% | 36.3 | 3.4 | 1.1% | 0.8% | 0.0% | 0.0% | 0.0% |
| DwtDctSvd | RoSteALS | - | parallel | - | 132 | 5.2% | 36.2 | 3.4 | 1.8% | 0.6% | 0.0% | 0.0% | 0.0% |
| DwtDctSvd | RoSteALS | - | series | 0.5 | 132 | 2.0% | 34.8 | 3.4 | 0.5% | 0.1% | 0.0% | 0.0% | 0.0% |
| DwtDctSvd | RoSteALS | - | series | - | 132 | 3.0% | 30.4 | 3.5 | 1.0% | 0.4% | 0.0% | 0.0% | 0.0% |
| DwtDctSvd | SSL (42dB,30bits) | - | series | 0.5 | 62 | 62.1% | 40.1 | 2.1 | 15.2% | 8.8% | 0.0% | 0.0% | 0.1% |
| DwtDctSvd | SSL (42dB,30bits) | - | series | - | 62 | 80.2% | 36.4 | 2.8 | 19.7% | 14.0% | 0.0% | 0.1% | 0.2% |
| DwtDctSvd | Trustmark B (0.95) | - | parallel | 0.5 | 132 | 4.9% | 41.5 | 2.5 | 1.1% | 0.9% | 0.0% | 0.0% | 0.0% |
| DwtDctSvd | Trustmark B (0.95) | - | parallel | - | 132 | 5.0% | 41.5 | 2.6 | 1.5% | 0.7% | 0.0% | 0.0% | 0.0% |

| First method | Second method | ECC | Ensemble | Clip | Capacity (bits) | Accuracy | Mean PSNR [dB] | Std PSNR [dB] | Robust. RivaGAN | Robust. SSL | Robust. TmLow | Robust. TmMed | Robust. TmHigh |
|---|---|---|---|---|---|---|---|---|---|---|---|---|---|
| DwtDctSvd | Trustmark B (0.95) | - | series | 0.5 | 132 | 26.9% | 39.7 | 2.5 | 6.7% | 3.7% | 0.0% | 0.0% | 0.0% |
| DwtDctSvd | Trustmark B (0.95) | - | series | - | 132 | 79.5% | 36.0 | 2.8 | 19.4% | 13.5% | 0.2% | 0.2% | 0.0% |
| DwtDctSvd | Trustmark Q (0.95) | - | parallel | 0.5 | 132 | 4.2% | 41.9 | 2.7 | 1.2% | 1.1% | 0.0% | 0.0% | 0.0% |
| DwtDctSvd | Trustmark Q (0.95) | - | parallel | - | 132 | 4.3% | 41.9 | 2.8 | 1.4% | 0.7% | 0.1% | 0.0% | 0.0% |
| DwtDctSvd | Trustmark Q (0.95) | - | series | 0.5 | 132 | 22.9% | 40.3 | 2.6 | 5.8% | 3.2% | 0.0% | 0.1% | 0.0% |
| DwtDctSvd | Trustmark Q (0.95) | - | series | - | 132 | 76.9% | 36.5 | 3.0 | 19.3% | 11.9% | 0.1% | 0.0% | 0.1% |
| HiDDeN | DwtDct | - | series | 0.5 | 80 | 1.0% | 35.5 | 3.5 | 0.1% | 0.1% | 0.0% | 0.0% | 0.0% |
| HiDDeN | DwtDct | - | series | - | 80 | 8.6% | 31.5 | 3.9 | 2.8% | 1.0% | 0.0% | 0.0% | 0.0% |
| HiDDeN | DwtDctSvd | - | series | 0.5 | 80 | 1.5% | 35.5 | 3.5 | 0.3% | 0.1% | 0.0% | 0.0% | 0.0% |
| HiDDeN | DwtDctSvd | - | series | - | 80 | 13.2% | 31.5 | 3.9 | 3.2% | 1.5% | 0.0% | 0.0% | 0.0% |
| HiDDeN | HiDDeN | - | parallel | 0.5 | 96 | 0.0% | 33.2 | 4.8 | 0.0% | 0.0% | 0.0% | 0.0% | 0.0% |
| HiDDeN | HiDDeN | - | parallel | - | 96 | 0.0% | 33.2 | 4.8 | 0.0% | 0.0% | 0.0% | 0.0% | 0.0% |
| HiDDeN | HiDDeN | - | series | 0.5 | 96 | 0.0% | 32.6 | 4.5 | 0.0% | 0.0% | 0.0% | 0.0% | 0.0% |
| HiDDeN | HiDDeN | - | series | - | 96 | 0.0% | 30.8 | 4.1 | 0.0% | 0.0% | 0.0% | 0.0% | 0.0% |
| HiDDeN | - | - | - | - | 48 | 15.0% | 32.7 | 4.5 | 5.3% | 1.9% | 0.1% | 0.0% | 0.1% |
| HiDDeN | RivaGAN | - | series | 0.5 | 80 | 0.3% | 36.7 | 2.3 | 0.0% | 0.0% | 0.0% | 0.0% | 0.0% |
| HiDDeN | RivaGAN | - | series | - | 80 | 9.4% | 31.8 | 3.8 | 2.4% | 1.2% | 0.0% | 0.0% | 0.1% |
| HiDDeN | RoSteALS | LC[148,49,34] | series | -0.2 | 49 | 81.6% | 30.2 | 3.8 | 38.8% | 16.5% | 0.2% | 0.1% | 0.2% |
| HiDDeN | RoSteALS | LC[148,49,34] | series | -0.1 | 49 | 80.5% | 30.4 | 3.8 | 40.1% | 18.1% | 0.2% | 0.2% | 0.0% |
| HiDDeN | RoSteALS | LC[148,49,34] | series | 0 | 49 | 79.2% | 30.7 | 3.8 | 35.5% | 14.3% | 0.1% | 0.1% | 0.1% |
| HiDDeN | RoSteALS | LC[148,49,34] | series | 0.1 | 49 | 78.3% | 31.0 | 3.8 | 37.9% | 15.1% | 0.0% | 0.0% | 0.0% |
| HiDDeN | RoSteALS | LC[148,49,34] | series | 0.2 | 49 | 77.6% | 31.2 | 3.8 | 35.9% | 15.1% | 0.0% | 0.2% | 0.2% |
| HiDDeN | RoSteALS | LC[148,49,34] | series | 0.3 | 49 | 75.6% | 31.5 | 3.8 | 35.0% | 14.0% | 0.0% | 0.1% | 0.1% |
| HiDDeN | RoSteALS | LC[148,49,34] | series | 0.4 | 49 | 72.3% | 31.7 | 3.8 | 34.1% | 14.3% | 0.1% | 0.1% | 0.1% |
| HiDDeN | RoSteALS | LC[148,49,34] | series | 0.5 | 49 | 70.6% | 32.0 | 3.8 | 33.6% | 12.8% | 0.1% | 0.1% | 0.0% |
| HiDDeN | RoSteALS | LC[148,49,34] | series | 0.6 | 49 | 68.5% | 32.3 | 3.8 | 31.3% | 12.3% | 0.0% | 0.0% | 0.0% |
| HiDDeN | RoSteALS | LC[148,49,34] | series | 0.7 | 49 | 66.7% | 32.5 | 3.8 | 30.3% | 12.8% | 0.1% | 0.2% | 0.1% |
| HiDDeN | RoSteALS | LC[148,49,34] | series | 0.8 | 49 | 65.0% | 32.8 | 3.9 | 29.2% | 10.7% | 0.1% | 0.2% | 0.2% |
| HiDDeN | RoSteALS | LC[148,49,34] | series | 0.9 | 49 | 62.9% | 33.0 | 4.0 | 29.1% | 11.9% | 0.2% | 0.1% | 0.0% |
| HiDDeN | RoSteALS | LC[148,49,34] | series | 1 | 49 | 61.0% | 33.3 | 4.0 | 27.9% | 10.7% | 0.0% | 0.0% | 0.0% |
| HiDDeN | RoSteALS | LC[148,49,34] | series | 1.1 | 49 | 58.8% | 33.5 | 4.1 | 27.9% | 10.8% | 0.1% | 0.0% | 0.0% |
| HiDDeN | RoSteALS | LC[148,49,34] | series | 1.2 | 49 | 56.9% | 33.8 | 4.2 | 26.7% | 9.7% | 0.0% | 0.0% | 0.1% |
| HiDDeN | RoSteALS | LC[148,49,34] | series | - | 49 | 85.6% | 28.8 | 3.8 | 40.8% | 16.4% | 0.1% | 0.3% | 0.3% |
| HiDDeN | RoSteALS | - | series | 0.5 | 148 | 0.0% | 32.0 | 3.8 | 0.0% | 0.0% | 0.0% | 0.0% | 0.0% |
| HiDDeN | RoSteALS | - | series | - | 148 | 1.8% | 28.8 | 3.8 | 1.0% | 0.1% | 0.0% | 0.0% | 0.0% |
| HiDDeN | SSL (42dB,30bits) | - | series | 0.5 | 78 | 0.1% | 37.3 | 2.3 | 0.0% | 0.0% | 0.0% | 0.0% | 0.0% |
| HiDDeN | SSL (42dB,30bits) | - | series | - | 78 | 14.4% | 32.0 | 3.9 | 4.7% | 1.4% | 0.1% | 0.0% | 0.0% |
| HiDDeN | Trustmark B (0.95) | - | parallel | 0.5 | 148 | 0.0% | 37.9 | 3.5 | 0.0% | 0.0% | 0.0% | 0.0% | 0.0% |
| HiDDeN | Trustmark B (0.95) | - | parallel | - | 148 | 0.0% | 37.7 | 3.9 | 0.0% | 0.0% | 0.0% | 0.0% | 0.0% |
| HiDDeN | Trustmark B (0.95) | - | series | 0.5 | 148 | 0.1% | 36.9 | 3.0 | 0.0% | 0.0% | 0.0% | 0.0% | 0.0% |
| HiDDeN | Trustmark B (0.95) | - | series | - | 148 | 12.5% | 31.9 | 4.0 | 4.2% | 1.7% | 0.0% | 0.0% | 0.0% |
| HiDDeN | Trustmark Q (0.95) | - | parallel | 0.5 | 148 | 0.0% | 38.2 | 3.5 | 0.0% | 0.0% | 0.0% | 0.0% | 0.0% |
| HiDDeN | Trustmark Q (0.95) | - | parallel | - | 148 | 0.0% | 37.9 | 4.0 | 0.0% | 0.0% | 0.0% | 0.0% | 0.0% |
| HiDDeN | Trustmark Q (0.95) | - | series | 0.5 | 148 | 0.0% | 37.5 | 3.1 | 0.0% | 0.0% | 0.0% | 0.0% | 0.0% |
| HiDDeN | Trustmark Q (0.95) | - | series | - | 148 | 10.4% | 32.1 | 4.2 | 3.2% | 1.3% | 0.1% | 0.0% | 0.0% |
| RivaGAN | DwtDct | - | series | 0.5 | 64 | 38.5% | 39.7 | 2.1 | 10.8% | 4.1% | 0.0% | 0.0% | 0.1% |
| RivaGAN | DwtDct | - | series | - | 64 | 43.3% | 36.0 | 2.6 | 11.0% | 5.8% | 0.1% | 0.0% | 0.0% |
| RivaGAN | DwtDctSvd | LC[64,36,12] | series | -0.2 | 36 | 85.0% | 37.3 | 3.4 | 20.5% | 15.3% | 0.4% | 0.2% | 0.2% |
| RivaGAN | DwtDctSvd | LC[64,36,12] | series | -0.1 | 36 | 84.5% | 37.5 | 3.4 | 19.9% | 17.6% | 0.1% | 0.2% | 0.2% |
| RivaGAN | DwtDctSvd | LC[64,36,12] | series | 0 | 36 | 84.2% | 37.8 | 3.3 | 19.6% | 16.1% | 0.3% | 0.1% | 0.0% |
| RivaGAN | DwtDctSvd | LC[64,36,12] | series | 0.1 | 36 | 82.8% | 38.2 | 3.0 | 19.6% | 15.2% | 0.2% | 0.2% | 0.2% |
| RivaGAN | DwtDctSvd | LC[64,36,12] | series | 0.2 | 36 | 81.5% | 38.5 | 2.7 | 20.1% | 15.0% | 0.1% | 0.1% | 0.1% |
| RivaGAN | DwtDctSvd | LC[64,36,12] | series | 0.3 | 36 | 78.7% | 39.0 | 2.5 | 21.2% | 15.4% | 0.2% | 0.4% | 0.1% |
| RivaGAN | DwtDctSvd | LC[64,36,12] | series | 0.4 | 36 | 77.1% | 39.3 | 2.3 | 18.6% | 13.5% | 0.2% | 0.2% | 0.2% |
| RivaGAN | DwtDctSvd | LC[64,36,12] | series | 0.5 | 36 | 74.8% | 39.6 | 2.1 | 18.7% | 13.6% | 0.1% | 0.1% | 0.1% |
| RivaGAN | DwtDctSvd | LC[64,36,12] | series | 0.6 | 36 | 72.7% | 39.9 | 1.9 | 18.5% | 13.9% | 0.3% | 0.1% | 0.1% |
| RivaGAN | DwtDctSvd | LC[64,36,12] | series | 0.7 | 36 | 70.1% | 40.2 | 1.7 | 16.2% | 11.3% | 0.3% | 0.2% | 0.2% |
| RivaGAN | DwtDctSvd | LC[64,36,12] | series | 0.8 | 36 | 67.1% | 40.6 | 1.8 | 16.5% | 12.5% | 0.1% | 0.1% | 0.0% |
| RivaGAN | DwtDctSvd | LC[64,36,12] | series | 0.9 | 36 | 64.0% | 40.9 | 1.8 | 15.1% | 10.6% | 0.1% | 0.1% | 0.2% |
| RivaGAN | DwtDctSvd | LC[64,36,12] | series | 1 | 36 | 61.3% | 41.3 | 1.8 | 15.4% | 11.0% | 0.1% | 0.2% | 0.0% |
| RivaGAN | DwtDctSvd | LC[64,36,12] | series | 1.1 | 36 | 58.7% | 41.6 | 1.9 | 16.3% | 10.1% | 0.1% | 0.2% | 0.1% |
| RivaGAN | DwtDctSvd | LC[64,36,12] | series | 1.2 | 36 | 55.9% | 42.0 | 2.0 | 13.1% | 10.5% | 0.1% | 0.1% | 0.1% |
| RivaGAN | DwtDctSvd | LC[64,36,12] | series | - | 36 | 86.9% | 36.0 | 2.6 | 21.2% | 16.2% | 0.2% | 0.1% | 0.0% |
| RivaGAN | DwtDctSvd | - | series | 0.5 | 64 | 57.3% | 39.6 | 2.1 | 13.9% | 8.8% | 0.0% | 0.0% | 0.0% |
| RivaGAN | DwtDctSvd | - | series | - | 64 | 69.9% | 36.0 | 2.6 | 17.2% | 11.8% | 0.1% | 0.1% | 0.0% |
| RivaGAN | HiDDeN | - | parallel | 0.5 | 80 | 0.2% | 37.9 | 3.1 | 0.0% | 0.0% | 0.0% | 0.0% | 0.0% |
| RivaGAN | HiDDeN | - | parallel | - | 80 | 0.2% | 37.6 | 3.7 | 0.1% | 0.0% | 0.0% | 0.0% | 0.0% |
| RivaGAN | HiDDeN | - | series | 0.5 | 80 | 0.5% | 36.7 | 2.3 | 0.1% | 0.1% | 0.0% | 0.0% | 0.0% |
| RivaGAN | HiDDeN | - | series | - | 80 | 11.9% | 31.8 | 3.8 | 3.7% | 1.5% | 0.0% | 0.0% | 0.0% |
| RivaGAN | - | - | - | - | 32 | 75.8% | 40.8 | 0.6 | 72.9% | 29.3% | 28.7% | 27.3% | 24.8% |
| RivaGAN | RivaGAN | - | parallel | 0.5 | 64 | 0.0% | 42.6 | 0.9 | 0.0% | 0.0% | 0.0% | 0.0% | 0.0% |
| RivaGAN | RivaGAN | - | parallel | - | 64 | 0.0% | 42.6 | 0.9 | 0.0% | 0.0% | 0.0% | 0.0% | 0.0% |
| RivaGAN | RivaGAN | - | series | 0.5 | 64 | 0.0% | 40.7 | 0.7 | 0.0% | 0.0% | 0.0% | 0.0% | 0.0% |
| RivaGAN | RivaGAN | - | series | - | 64 | 0.0% | 37.8 | 1.2 | 0.0% | 0.0% | 0.0% | 0.0% | 0.0% |
| RivaGAN | RoSteALS | LC[131,32,37] | parallel | -0.2 | 32 | 97.4% | 36.4 | 3.0 | 51.6% | 28.4% | 0.2% | 0.0% | 0.2% |
| RivaGAN | RoSteALS | LC[131,32,37] | parallel | -0.1 | 32 | 97.4% | 36.4 | 3.0 | 48.9% | 28.7% | 0.4% | 0.4% | 0.4% |
| RivaGAN | RoSteALS | LC[131,32,37] | parallel | 0 | 32 | 97.4% | 36.4 | 3.0 | 52.3% | 27.7% | 0.4% | 0.4% | 0.4% |
| RivaGAN | RoSteALS | LC[131,32,37] | parallel | 0.1 | 32 | 97.4% | 36.4 | 3.0 | 48.3% | 29.3% | 0.3% | 0.7% | 0.2% |
| RivaGAN | RoSteALS | LC[131,32,37] | parallel | 0.2 | 32 | 97.4% | 36.4 | 3.0 | 48.2% | 27.8% | 0.3% | 0.3% | 0.4% |
| RivaGAN | RoSteALS | LC[131,32,37] | parallel | 0.3 | 32 | 97.4% | 36.4 | 3.0 | 51.4% | 29.2% | 0.3% | 0.2% | 0.3% |
| RivaGAN | RoSteALS | LC[131,32,37] | parallel | 0.4 | 32 | 97.0% | 36.5 | 2.9 | 49.0% | 28.3% | 0.6% | 0.2% | 0.1% |
| RivaGAN | RoSteALS | LC[131,32,37] | parallel | 0.5 | 32 | 94.9% | 36.8 | 2.5 | 49.1% | 29.0% | 0.1% | 0.5% | 0.2% |
| RivaGAN | RoSteALS | LC[131,32,37] | parallel | 0.6 | 32 | 90.7% | 37.3 | 2.0 | 46.8% | 25.7% | 0.5% | 0.2% | 0.3% |
| RivaGAN | RoSteALS | LC[131,32,37] | parallel | 0.7 | 32 | 85.0% | 38.1 | 1.5 | 42.9% | 22.5% | 0.3% | 0.2% | 0.2% |
| RivaGAN | RoSteALS | LC[131,32,37] | parallel | 0.8 | 32 | 79.8% | 39.0 | 1.1 | 40.6% | 21.9% | 0.4% | 0.3% | 0.1% |
| RivaGAN | RoSteALS | LC[131,32,37] | parallel | 0.9 | 32 | 74.5% | 39.9 | 0.9 | 37.2% | 20.0% | 0.1% | 0.0% | 0.2% |

| First method | Second method | ECC | Ensemble | Clip | Capacity (bits) | Accuracy | Mean PSNR [dB] | Std PSNR [dB] | Robust. RivaGAN | Robust. SSL | Robust. TmLow | Robust. TmMed | Robust. TmHigh |
|---|---|---|---|---|---|---|---|---|---|---|---|---|---|
| RivaGAN | RoSteALS | LC[131,32,37] | parallel | 1 | 32 | 67.2% | 40.8 | 0.7 | 32.4% | 19.4% | 0.1% | 0.3% | 0.2% |
| RivaGAN | RoSteALS | LC[131,32,37] | parallel | 1.1 | 32 | 59.6% | 41.8 | 0.8 | 30.2% | 16.8% | 0.0% | 0.1% | 0.1% |
| RivaGAN | RoSteALS | LC[131,32,37] | parallel | 1.2 | 32 | 53.1% | 42.7 | 1.0 | 26.4% | 15.4% | 0.3% | 0.1% | 0.0% |
| RivaGAN | RoSteALS | LC[131,32,37] | parallel | - | 32 | 97.4% | 36.4 | 3.0 | 48.7% | 29.3% | 0.2% | 0.3% | 0.5% |
| RivaGAN | RoSteALS | LC[131,32,37] | series | -0.2 | 32 | 99.5% | 31.0 | 3.4 | 51.1% | 31.0% | 0.6% | 0.5% | 0.8% |
| RivaGAN | RoSteALS | LC[131,32,37] | series | -0.1 | 32 | 99.5% | 31.0 | 3.5 | 51.5% | 31.3% | 0.2% | 0.4% | 0.4% |
| RivaGAN | RoSteALS | LC[131,32,37] | series | 0 | 32 | 99.5% | 31.3 | 3.6 | 50.9% | 30.9% | 0.4% | 0.5% | 0.2% |
| RivaGAN | RoSteALS | LC[131,32,37] | series | 0.1 | 32 | 99.1% | 32.3 | 3.2 | 49.9% | 32.1% | 0.5% | 0.5% | 0.2% |
| RivaGAN | RoSteALS | LC[131,32,37] | series | 0.2 | 32 | 98.5% | 33.2 | 2.9 | 51.2% | 31.0% | 0.2% | 0.3% | 0.4% |
| RivaGAN | RoSteALS | LC[131,32,37] | series | 0.3 | 32 | 96.9% | 34.2 | 2.6 | 49.4% | 28.7% | 0.4% | 0.4% | 0.3% |
| RivaGAN | RoSteALS | LC[131,32,37] | series | 0.4 | 32 | 95.0% | 35.1 | 2.2 | 47.2% | 27.3% | 0.4% | 0.3% | 0.3% |
| RivaGAN | RoSteALS | LC[131,32,37] | series | 0.5 | 32 | 91.7% | 36.1 | 1.9 | 47.7% | 25.4% | 0.2% | 0.3% | 0.3% |
| RivaGAN | RoSteALS | LC[131,32,37] | series | 0.6 | 32 | 87.6% | 37.0 | 1.6 | 44.7% | 24.4% | 0.3% | 0.4% | 0.3% |
| RivaGAN | RoSteALS | LC[131,32,37] | series | 0.7 | 32 | 82.9% | 38.0 | 1.2 | 43.0% | 22.9% | 0.2% | 0.2% | 0.3% |
| RivaGAN | RoSteALS | LC[131,32,37] | series | 0.8 | 32 | 77.2% | 38.9 | 1.0 | 38.4% | 21.0% | 0.3% | 0.1% | 0.3% |
| RivaGAN | RoSteALS | LC[131,32,37] | series | 0.9 | 32 | 71.7% | 39.9 | 0.8 | 35.2% | 19.3% | 0.2% | 0.2% | 0.3% |
| RivaGAN | RoSteALS | LC[131,32,37] | series | 1 | 32 | 65.3% | 40.8 | 0.7 | 32.5% | 16.5% | 0.2% | 0.0% | 0.1% |
| RivaGAN | RoSteALS | LC[131,32,37] | series | 1.1 | 32 | 58.1% | 41.7 | 0.7 | 27.1% | 16.0% | 0.1% | 0.2% | 0.1% |
| RivaGAN | RoSteALS | LC[131,32,37] | series | 1.2 | 32 | 51.6% | 42.7 | 0.9 | 24.8% | 13.1% | 0.1% | 0.0% | 0.1% |
| RivaGAN | RoSteALS | LC[131,32,37] | series | - | 32 | 99.4% | 30.9 | 3.3 | 51.4% | 29.8% | 0.4% | 0.3% | 0.6% |
| RivaGAN | RoSteALS | - | parallel | 0.5 | 132 | 4.3% | 36.8 | 2.5 | 2.3% | 0.7% | 0.0% | 0.0% | 0.0% |
| RivaGAN | RoSteALS | - | parallel | - | 132 | 4.3% | 36.4 | 3.0 | 2.1% | 0.9% | 0.0% | 0.0% | 0.0% |
| RivaGAN | RoSteALS | - | series | 0.5 | 132 | 5.9% | 36.1 | 1.9 | 2.4% | 0.7% | 0.0% | 0.0% | 0.0% |
| RivaGAN | RoSteALS | - | series | - | 132 | 24.0% | 30.9 | 3.3 | 10.7% | 5.6% | 0.0% | 0.0% | 0.0% |
| RivaGAN | SSL (42dB,30bits) | - | series | 0.5 | 62 | 63.7% | 41.4 | 0.6 | 50.9% | 20.7% | 13.9% | 11.8% | 10.4% |
| RivaGAN | SSL (42dB,30bits) | - | series | - | 62 | 69.9% | 38.4 | 0.6 | 64.8% | 24.7% | 21.0% | 19.1% | 15.3% |
| RivaGAN | Trustmark B (0.95) | - | parallel | 0.5 | 132 | 2.9% | 42.8 | 1.2 | 4.3% | 0.7% | 1.0% | 1.4% | 0.8% |
| RivaGAN | Trustmark B (0.95) | - | parallel | - | 132 | 2.9% | 42.8 | 1.2 | 4.3% | 0.7% | 1.0% | 1.0% | 1.1% |
| RivaGAN | Trustmark B (0.95) | - | series | 0.5 | 132 | 26.3% | 40.9 | 1.1 | 28.4% | 5.7% | 9.8% | 9.0% | 8.1% |
| RivaGAN | Trustmark B (0.95) | - | series | - | 132 | 68.0% | 37.5 | 1.1 | 64.8% | 18.9% | 22.9% | 21.4% | 18.0% |
| RivaGAN | Trustmark Q (0.95) | - | parallel | 0.5 | 132 | 3.5% | 43.3 | 1.2 | 4.1% | 0.6% | 1.1% | 0.8% | 0.6% |
| RivaGAN | Trustmark Q (0.95) | - | parallel | - | 132 | 3.5% | 43.3 | 1.2 | 4.2% | 0.8% | 0.7% | 0.6% | 0.4% |
| RivaGAN | Trustmark Q (0.95) | - | series | 0.5 | 132 | 21.6% | 41.5 | 1.2 | 24.1% | 5.1% | 7.9% | 6.6% | 6.0% |
| RivaGAN | Trustmark Q (0.95) | - | series | - | 132 | 67.2% | 38.0 | 1.2 | 65.0% | 16.3% | 22.4% | 20.9% | 19.8% |
| RoSteALS | DwtDct | - | series | 0.5 | 132 | 13.4% | 34.8 | 3.3 | 3.7% | 1.1% | 0.0% | 0.0% | 0.0% |
| RoSteALS | DwtDct | - | series | - | 132 | 36.8% | 30.4 | 3.4 | 9.7% | 4.6% | 0.0% | 0.0% | 0.0% |
| RoSteALS | DwtDctSvd | - | series | 0.5 | 132 | 16.0% | 34.8 | 3.4 | 4.2% | 2.7% | 0.0% | 0.0% | 0.0% |
| RoSteALS | DwtDctSvd | - | series | - | 132 | 65.1% | 30.4 | 3.4 | 16.1% | 11.3% | 0.0% | 0.1% | 0.0% |
| RoSteALS | HiDDeN | LC[148,102,14] | series | 1 | 102 | 21.5% | 33.3 | 4.0 | 8.2% | 3.2% | 0.0% | 0.0% | 0.0% |
| RoSteALS | HiDDeN | LC[148,49,34] | parallel | -0.2 | 49 | 60.0% | 34.5 | 3.7 | 28.2% | 10.5% | 0.1% | 0.0% | 0.0% |
| RoSteALS | HiDDeN | LC[148,49,34] | parallel | -0.1 | 49 | 60.0% | 34.5 | 3.7 | 28.1% | 10.9% | 0.1% | 0.0% | 0.0% |
| RoSteALS | HiDDeN | LC[148,49,34] | parallel | 0 | 49 | 60.0% | 34.5 | 3.7 | 26.4% | 10.2% | 0.0% | 0.0% | 0.0% |
| RoSteALS | HiDDeN | LC[148,49,34] | parallel | 0.1 | 49 | 60.0% | 34.5 | 3.7 | 26.2% | 10.6% | 0.1% | 0.0% | 0.0% |
| RoSteALS | HiDDeN | LC[148,49,34] | parallel | 0.2 | 49 | 60.0% | 34.5 | 3.7 | 25.4% | 10.6% | 0.0% | 0.0% | 0.0% |
| RoSteALS | HiDDeN | LC[148,49,34] | parallel | 0.3 | 49 | 60.0% | 34.5 | 3.7 | 27.1% | 9.6% | 0.0% | 0.1% | 0.0% |
| RoSteALS | HiDDeN | LC[148,49,34] | parallel | 0.4 | 49 | 60.0% | 34.5 | 3.7 | 27.0% | 11.7% | 0.0% | 0.0% | 0.1% |
| RoSteALS | HiDDeN | LC[148,49,34] | parallel | 0.5 | 49 | 60.0% | 34.5 | 3.7 | 25.6% | 10.1% | 0.1% | 0.0% | 0.1% |
| RoSteALS | HiDDeN | LC[148,49,34] | parallel | 0.6 | 49 | 60.0% | 34.5 | 3.7 | 26.4% | 11.7% | 0.0% | 0.0% | 0.0% |
| RoSteALS | HiDDeN | LC[148,49,34] | parallel | 0.7 | 49 | 59.9% | 34.5 | 3.7 | 25.4% | 10.4% | 0.0% | 0.1% | 0.0% |
| RoSteALS | HiDDeN | LC[148,49,34] | parallel | 0.8 | 49 | 59.9% | 34.5 | 3.7 | 25.6% | 10.0% | 0.2% | 0.1% | 0.0% |
| RoSteALS | HiDDeN | LC[148,49,34] | parallel | 0.9 | 49 | 59.4% | 34.6 | 3.7 | 27.2% | 10.8% | 0.0% | 0.0% | 0.0% |
| RoSteALS | HiDDeN | LC[148,49,34] | parallel | 1 | 49 | 58.6% | 34.7 | 3.7 | 25.3% | 10.1% | 0.0% | 0.1% | 0.0% |
| RoSteALS | HiDDeN | LC[148,49,34] | parallel | 1.1 | 49 | 58.0% | 34.8 | 3.8 | 25.0% | 10.4% | 0.0% | 0.0% | 0.1% |
| RoSteALS | HiDDeN | LC[148,49,34] | parallel | 1.2 | 49 | 57.3% | 34.9 | 3.8 | 25.3% | 9.0% | 0.0% | 0.0% | 0.1% |
| RoSteALS | HiDDeN | LC[148,49,34] | parallel | - | 49 | 60.0% | 34.5 | 3.7 | 25.7% | 11.6% | 0.0% | 0.0% | 0.0% |
| RoSteALS | HiDDeN | LC[148,49,34] | series | -0.2 | 49 | 91.9% | 30.2 | 3.8 | 45.6% | 20.2% | 0.2% | 0.2% | 0.0% |
| RoSteALS | HiDDeN | LC[148,49,34] | series | -0.1 | 49 | 91.0% | 30.4 | 3.8 | 44.2% | 19.1% | 0.1% | 0.2% | 0.0% |
| RoSteALS | HiDDeN | LC[148,49,34] | series | 0 | 49 | 89.9% | 30.7 | 3.8 | 43.2% | 19.2% | 0.3% | 0.2% | 0.1% |
| RoSteALS | HiDDeN | LC[148,49,34] | series | 0.1 | 49 | 89.4% | 31.0 | 3.8 | 41.0% | 21.6% | 0.1% | 0.1% | 0.1% |
| RoSteALS | HiDDeN | LC[148,49,34] | series | 0.2 | 49 | 88.3% | 31.2 | 3.7 | 44.1% | 19.5% | 0.2% | 0.1% | 0.3% |
| RoSteALS | HiDDeN | LC[148,49,34] | series | 0.3 | 49 | 87.0% | 31.5 | 3.8 | 41.2% | 19.4% | 0.1% | 0.1% | 0.0% |
| RoSteALS | HiDDeN | LC[148,49,34] | series | 0.4 | 49 | 85.1% | 31.7 | 3.8 | 42.4% | 17.7% | 0.1% | 0.0% | 0.2% |
| RoSteALS | HiDDeN | LC[148,49,34] | series | 0.5 | 49 | 81.6% | 32.0 | 3.8 | 38.9% | 17.6% | 0.1% | 0.2% | 0.1% |
| RoSteALS | HiDDeN | LC[148,49,34] | series | 0.6 | 49 | 79.7% | 32.3 | 3.8 | 37.8% | 16.1% | 0.1% | 0.1% | 0.0% |
| RoSteALS | HiDDeN | LC[148,49,34] | series | 0.7 | 49 | 78.4% | 32.5 | 3.8 | 37.0% | 15.5% | 0.2% | 0.1% | 0.0% |
| RoSteALS | HiDDeN | LC[148,49,34] | series | 0.8 | 49 | 75.6% | 32.8 | 3.9 | 36.8% | 15.4% | 0.0% | 0.1% | 0.2% |
| RoSteALS | HiDDeN | LC[148,49,34] | series | 0.9 | 49 | 73.3% | 33.0 | 4.0 | 34.7% | 14.6% | 0.0% | 0.0% | 0.2% |
| RoSteALS | HiDDeN | LC[148,49,34] | series | 1 | 49 | 72.2% | 33.3 | 4.0 | 34.5% | 14.4% | 0.2% | 0.0% | 0.0% |
| RoSteALS | HiDDeN | LC[148,49,34] | series | 1.1 | 49 | 69.4% | 33.5 | 4.1 | 35.1% | 13.1% | 0.1% | 0.2% | 0.0% |
| RoSteALS | HiDDeN | LC[148,49,34] | series | 1.2 | 49 | 66.8% | 33.8 | 4.2 | 30.1% | 12.7% | 0.1% | 0.0% | 0.0% |
| RoSteALS | HiDDeN | LC[148,49,34] | series | - | 49 | 94.9% | 28.7 | 3.8 | 47.1% | 24.5% | 0.2% | 0.3% | 0.2% |
| RoSteALS | HiDDeN | - | parallel | 0.5 | 148 | 0.0% | 34.5 | 3.7 | 0.0% | 0.0% | 0.0% | 0.0% | 0.0% |
| RoSteALS | HiDDeN | - | parallel | - | 148 | 0.0% | 34.5 | 3.7 | 0.0% | 0.0% | 0.0% | 0.0% | 0.0% |
| RoSteALS | HiDDeN | - | series | 0.5 | 148 | 0.2% | 32.0 | 3.8 | 0.0% | 0.0% | 0.0% | 0.0% | 0.0% |
| RoSteALS | HiDDeN | - | series | - | 148 | 7.0% | 28.7 | 3.8 | 1.9% | 1.1% | 0.0% | 0.0% | 0.0% |
| RoSteALS | - | LC[100,32,25] | - | - | 32 | 99.9% | 31.4 | 3.6 | 50.4% | 31.7% | 0.4% | 0.5% | 0.5% |
| RoSteALS | - | LC[100,48,20] | - | - | 48 | 99.9% | 31.4 | 3.6 | 50.0% | 31.6% | 0.3% | 0.4% | 0.4% |
| RoSteALS | - | - | - | - | 100 | 72.1% | 31.4 | 3.6 | 35.4% | 22.1% | 0.1% | 0.1% | 0.1% |
| RoSteALS | RivaGAN | LC[131,32,37] | series | -0.2 | 32 | 99.9% | 30.8 | 3.4 | 50.0% | 30.6% | 0.5% | 0.8% | 0.5% |
| RoSteALS | RivaGAN | LC[131,32,37] | series | -0.1 | 32 | 99.8% | 30.9 | 3.5 | 52.2% | 30.5% | 0.4% | 0.7% | 0.5% |
| RoSteALS | RivaGAN | LC[131,32,37] | series | 0 | 32 | 99.9% | 31.3 | 3.6 | 51.7% | 31.2% | 0.5% | 0.4% | 0.3% |
| RoSteALS | RivaGAN | LC[131,32,37] | series | 0.1 | 32 | 99.8% | 32.3 | 3.3 | 51.8% | 32.0% | 0.4% | 0.4% | 0.6% |
| RoSteALS | RivaGAN | LC[131,32,37] | series | 0.2 | 32 | 99.5% | 33.2 | 2.9 | 51.9% | 31.2% | 0.6% | 0.3% | 0.3% |
| RoSteALS | RivaGAN | LC[131,32,37] | series | 0.3 | 32 | 98.8% | 34.2 | 2.6 | 50.0% | 30.8% | 0.5% | 0.4% | 0.2% |
| RoSteALS | RivaGAN | LC[131,32,37] | series | 0.4 | 32 | 97.1% | 35.1 | 2.2 | 49.2% | 29.6% | 0.3% | 0.3% | 0.4% |
| RoSteALS | RivaGAN | LC[131,32,37] | series | 0.5 | 32 | 94.4% | 36.1 | 1.9 | 48.9% | 27.5% | 0.3% | 0.3% | 0.1% |

| First method | Second method | ECC | Ensemble | Clip | Capacity (bits) | Accuracy | Mean PSNR [dB] | Std PSNR [dB] | Robust. RivaGAN | Robust. SSL | Robust. TmLow | Robust. TmMed | Robust. TmHigh |
|---|---|---|---|---|---|---|---|---|---|---|---|---|---|
| RoSteALS | RivaGAN | LC[131,32,37] | series | 0.6 | 32 | 88.7% | 37.0 | 1.6 | 43.0% | 24.3% | 0.0% | 0.2% | 0.1% |
| RoSteALS | RivaGAN | LC[131,32,37] | series | 0.7 | 32 | 83.2% | 38.0 | 1.2 | 43.3% | 22.1% | 0.1% | 0.1% | 0.1% |
| RoSteALS | RivaGAN | LC[131,32,37] | series | 0.8 | 32 | 77.9% | 38.9 | 1.0 | 40.1% | 22.3% | 0.1% | 0.2% | 0.1% |
| RoSteALS | RivaGAN | LC[131,32,37] | series | 0.9 | 32 | 71.4% | 39.9 | 0.8 | 37.4% | 19.4% | 0.0% | 0.1% | 0.2% |
| RoSteALS | RivaGAN | LC[131,32,37] | series | 1 | 32 | 64.7% | 40.8 | 0.7 | 33.7% | 16.8% | 0.1% | 0.1% | 0.1% |
| RoSteALS | RivaGAN | LC[131,32,37] | series | 1.1 | 32 | 55.3% | 41.8 | 0.7 | 27.0% | 15.8% | 0.2% | 0.0% | 0.1% |
| RoSteALS | RivaGAN | LC[131,32,37] | series | 1.2 | 32 | 49.2% | 42.7 | 0.9 | 23.9% | 14.8% | 0.1% | 0.2% | 0.1% |
| RoSteALS | RivaGAN | LC[131,32,37] | series | - | 32 | 99.9% | 30.7 | 3.1 | 54.6% | 28.3% | 0.5% | 0.4% | 0.5% |
| RoSteALS | RivaGAN | - | series | 0.5 | 132 | 8.0% | 36.1 | 1.9 | 4.0% | 1.8% | 0.0% | 0.0% | 0.0% |
| RoSteALS | RivaGAN | - | series | - | 132 | 42.9% | 30.7 | 3.1 | 22.5% | 9.3% | 0.0% | 0.0% | 0.0% |
| RoSteALS | RoSteALS | - | parallel | 0.5 | 200 | 0.0% | 32.2 | 3.6 | 0.0% | 0.0% | 0.0% | 0.0% | 0.0% |
| RoSteALS | RoSteALS | - | parallel | - | 200 | 0.0% | 32.2 | 3.6 | 0.0% | 0.0% | 0.0% | 0.0% | 0.0% |
| RoSteALS | RoSteALS | - | series | 0.5 | 200 | 0.0% | 31.3 | 3.6 | 0.0% | 0.0% | 0.0% | 0.0% | 0.0% |
| RoSteALS | RoSteALS | - | series | - | 200 | 0.0% | 27.8 | 3.2 | 0.0% | 0.0% | 0.0% | 0.0% | 0.0% |
| RoSteALS | SSL (42dB,30bits) | LC[130,100,10] | series | 0.2 | 100 | 86.8% | 33.5 | 2.9 | 43.8% | 21.6% | 0.4% | 0.1% | 0.0% |
| RoSteALS | SSL (42dB,30bits) | - | series | 0.5 | 130 | 20.0% | 36.7 | 1.9 | 8.3% | 4.0% | 0.0% | 0.1% | 0.0% |
| RoSteALS | SSL (42dB,30bits) | - | series | - | 130 | 66.8% | 30.9 | 3.3 | 32.5% | 16.6% | 0.0% | 0.1% | 0.1% |
| RoSteALS | Trustmark B (0.95) | - | parallel | 0.5 | 200 | 4.6% | 36.6 | 3.0 | 1.8% | 1.0% | 0.0% | 0.0% | 0.0% |
| RoSteALS | Trustmark B (0.95) | - | parallel | - | 200 | 4.7% | 36.4 | 3.2 | 2.7% | 1.6% | 0.0% | 0.0% | 0.0% |
| RoSteALS | Trustmark B (0.95) | - | series | 0.5 | 200 | 5.8% | 36.3 | 2.7 | 2.6% | 1.9% | 0.0% | 0.0% | 0.0% |
| RoSteALS | Trustmark B (0.95) | - | series | - | 200 | 48.5% | 30.8 | 3.3 | 25.7% | 14.7% | 0.0% | 0.1% | 0.0% |
| RoSteALS | Trustmark Q (0.95) | - | parallel | 0.5 | 200 | 2.5% | 36.9 | 2.9 | 1.2% | 0.6% | 0.0% | 0.0% | 0.0% |
| RoSteALS | Trustmark Q (0.95) | - | parallel | - | 200 | 3.0% | 36.5 | 3.2 | 1.6% | 0.8% | 0.0% | 0.0% | 0.0% |
| RoSteALS | Trustmark Q (0.95) | - | series | 0.5 | 200 | 5.3% | 36.8 | 2.8 | 3.1% | 1.2% | 0.0% | 0.0% | 0.0% |
| RoSteALS | Trustmark Q (0.95) | - | series | - | 200 | 50.2% | 30.8 | 3.4 | 24.9% | 16.5% | 0.0% | 0.2% | 0.0% |
| SSL (42dB,30bits) | DwtDct | - | parallel | 0.5 | 62 | 9.4% | 42.3 | 2.6 | 2.7% | 0.9% | 0.0% | 0.0% | 0.0% |
| SSL (42dB,30bits) | DwtDct | - | parallel | - | 62 | 9.4% | 42.3 | 2.7 | 2.3% | 0.9% | 0.0% | 0.0% | 0.0% |
| SSL (42dB,30bits) | DwtDct | - | series | 0.5 | 62 | 25.6% | 40.2 | 2.1 | 7.1% | 3.3% | 0.0% | 0.0% | 0.0% |
| SSL (42dB,30bits) | DwtDct | - | series | - | 62 | 32.0% | 36.5 | 2.8 | 7.8% | 2.8% | 0.0% | 0.0% | 0.0% |
| SSL (42dB,30bits) | DwtDctSvd | - | parallel | 0.5 | 62 | 21.5% | 42.2 | 2.5 | 6.1% | 2.2% | 0.0% | 0.0% | 0.0% |
| SSL (42dB,30bits) | DwtDctSvd | - | parallel | - | 62 | 21.5% | 42.1 | 2.7 | 5.1% | 1.9% | 0.0% | 0.0% | 0.0% |
| SSL (42dB,30bits) | DwtDctSvd | - | series | 0.5 | 62 | 46.8% | 40.2 | 2.1 | 12.3% | 6.9% | 0.0% | 0.0% | 0.1% |
| SSL (42dB,30bits) | DwtDctSvd | - | series | - | 62 | 64.0% | 36.4 | 2.8 | 15.4% | 9.2% | 0.0% | 0.1% | 0.0% |
| SSL (42dB,30bits) | HiDDeN | - | parallel | 0.5 | 78 | 0.1% | 38.3 | 3.1 | 0.0% | 0.0% | 0.0% | 0.0% | 0.0% |
| SSL (42dB,30bits) | HiDDeN | - | parallel | - | 78 | 0.3% | 37.9 | 3.8 | 0.0% | 0.0% | 0.0% | 0.0% | 0.0% |
| SSL (42dB,30bits) | HiDDeN | - | series | 0.5 | 78 | 0.1% | 37.4 | 2.3 | 0.0% | 0.0% | 0.0% | 0.0% | 0.0% |
| SSL (42dB,30bits) | HiDDeN | - | series | - | 78 | 5.1% | 32.0 | 3.9 | 1.4% | 0.6% | 0.0% | 0.0% | 0.0% |
| SSL (42dB,30bits) | - | - | - | - | 30 | 93.8% | 42.1 | 0.6 | 86.1% | 75.8% | 58.0% | 52.2% | 47.9% |
| SSL (42dB,30bits) | RivaGAN | - | parallel | 0.5 | 62 | 27.8% | 43.7 | 0.5 | 11.6% | 4.3% | 1.5% | 1.0% | 1.1% |
| SSL (42dB,30bits) | RivaGAN | - | parallel | - | 62 | 27.8% | 43.7 | 0.5 | 10.7% | 4.3% | 0.9% | 1.0% | 0.4% |
| SSL (42dB,30bits) | RivaGAN | - | series | 0.5 | 62 | 53.4% | 41.5 | 0.6 | 32.8% | 12.8% | 6.1% | 4.2% | 3.7% |
| SSL (42dB,30bits) | RivaGAN | - | series | - | 62 | 64.5% | 38.4 | 0.5 | 51.4% | 17.5% | 11.7% | 10.8% | 8.2% |
| SSL (42dB,30bits) | RoSteALS | - | parallel | 0.5 | 130 | 8.0% | 37.3 | 2.5 | 2.7% | 1.4% | 0.0% | 0.0% | 0.0% |
| SSL (42dB,30bits) | RoSteALS | - | parallel | - | 130 | 8.2% | 36.8 | 3.2 | 3.1% | 1.2% | 0.0% | 0.0% | 0.0% |
| SSL (42dB,30bits) | RoSteALS | - | series | 0.5 | 130 | 2.3% | 36.7 | 1.9 | 0.7% | 0.4% | 0.0% | 0.0% | 0.0% |
| SSL (42dB,30bits) | RoSteALS | - | series | - | 130 | 15.6% | 30.9 | 3.3 | 5.7% | 2.2% | 0.0% | 0.0% | 0.0% |
| SSL (42dB,30bits) | SSL (42dB,30bits) | - | parallel | 0.5 | 60 | 0.0% | 44.2 | 0.6 | 0.0% | 0.0% | 0.0% | 0.0% | 0.0% |
| SSL (42dB,30bits) | SSL (42dB,30bits) | - | parallel | - | 60 | 0.0% | 44.2 | 0.6 | 0.0% | 0.0% | 0.0% | 0.0% | 0.0% |
| SSL (42dB,30bits) | SSL (42dB,30bits) | - | series | 0.5 | 60 | 0.0% | 42.1 | 0.6 | 0.0% | 0.0% | 0.0% | 0.0% | 0.0% |
| SSL (42dB,30bits) | SSL (42dB,30bits) | - | series | 0.5 | 60 | 0.0% | 42.1 | 0.6 | 0.0% | 0.0% | 0.0% | 0.0% | 0.0% |
| SSL (42dB,30bits) | SSL (42dB,30bits) | - | series | - | 60 | 0.0% | 39.1 | 0.6 | 0.0% | 0.0% | 0.0% | 0.0% | 0.0% |
| SSL (42dB,30bits) | SSL (42dB,30bits) | - | series | - | 60 | 0.0% | 39.1 | 0.6 | 0.0% | 0.0% | 0.0% | 0.0% | 0.0% |
| SSL (42dB,30bits) | Trustmark B (0.95) | - | parallel | 0.5 | 130 | 5.3% | 43.8 | 1.1 | 2.8% | 1.6% | 1.2% | 0.8% | 0.5% |
| SSL (42dB,30bits) | Trustmark B (0.95) | - | parallel | - | 130 | 5.3% | 43.8 | 1.1 | 3.1% | 1.6% | 1.4% | 0.8% | 0.7% |
| SSL (42dB,30bits) | Trustmark B (0.95) | - | series | 0.5 | 130 | 35.4% | 41.7 | 1.0 | 28.2% | 16.3% | 13.8% | 12.3% | 9.5% |
| SSL (42dB,30bits) | Trustmark B (0.95) | - | series | - | 130 | 77.7% | 38.4 | 1.2 | 68.8% | 41.5% | 43.0% | 36.7% | 32.7% |
| SSL (42dB,30bits) | Trustmark Q (0.95) | - | parallel | 0.5 | 130 | 2.5% | 44.3 | 1.1 | 2.0% | 1.2% | 0.5% | 0.5% | 0.4% |
| SSL (42dB,30bits) | Trustmark Q (0.95) | - | parallel | - | 130 | 2.5% | 44.3 | 1.1 | 2.1% | 1.0% | 0.4% | 0.5% | 0.5% |
| SSL (42dB,30bits) | Trustmark Q (0.95) | - | series | 0.5 | 130 | 35.2% | 42.3 | 1.2 | 28.3% | 14.2% | 13.4% | 11.1% | 9.0% |
| SSL (42dB,30bits) | Trustmark Q (0.95) | - | series | - | 130 | 77.3% | 39.0 | 1.2 | 66.8% | 37.6% | 40.5% | 34.9% | 31.8% |
| Trustmark B (0.95) | DwtDct | - | series | 0.5 | 132 | 20.5% | 39.7 | 2.4 | 4.7% | 2.5% | 0.0% | 0.0% | 0.0% |
| Trustmark B (0.95) | DwtDct | - | series | - | 132 | 49.3% | 36.2 | 2.8 | 11.3% | 5.4% | 0.1% | 0.0% | 0.0% |
| Trustmark B (0.95) | DwtDctSvd | - | series | 0.5 | 132 | 26.7% | 39.7 | 2.5 | 6.4% | 4.2% | 0.1% | 0.0% | 0.0% |
| Trustmark B (0.95) | DwtDctSvd | - | series | - | 132 | 88.5% | 36.0 | 2.8 | 23.3% | 16.4% | 0.3% | 0.1% | 0.4% |
| Trustmark B (0.95) | HiDDeN | - | series | 0.5 | 148 | 0.0% | 36.9 | 3.0 | 0.0% | 0.0% | 0.0% | 0.0% | 0.0% |
| Trustmark B (0.95) | HiDDeN | - | series | - | 148 | 13.2% | 31.9 | 4.0 | 4.8% | 1.4% | 0.0% | 0.0% | 0.1% |
| Trustmark B (0.95) | - | - | - | - | 100 | 96.5% | 41.3 | 2.1 | 96.9% | 67.9% | 85.2% | 83.1% | 79.4% |
| Trustmark B (0.95) | RivaGAN | - | series | 0.5 | 132 | 24.3% | 40.9 | 1.1 | 27.2% | 6.2% | 9.4% | 8.5% | 6.6% |
| Trustmark B (0.95) | RivaGAN | - | series | - | 132 | 73.9% | 37.5 | 1.1 | 72.6% | 20.1% | 26.7% | 26.7% | 21.4% |
| Trustmark B (0.95) | RoSteALS | - | series | 0.5 | 200 | 4.2% | 36.2 | 2.7 | 1.9% | 1.1% | 0.0% | 0.0% | 0.0% |
| Trustmark B (0.95) | RoSteALS | - | series | - | 200 | 31.1% | 30.7 | 3.3 | 15.6% | 9.1% | 0.0% | 0.0% | 0.0% |
| Trustmark B (0.95) | SSL (42dB,100bits) | LC[200,100,32] | series | 1 | 100 | 80.6% | 42.5 | 1.1 | 67.4% | 39.3% | 40.8% | 37.9% | 32.9% |
| Trustmark B (0.95) | SSL (42dB,100bits) | - | series | 1 | 200 | 22.9% | 42.5 | 1.1 | 11.4% | 6.2% | 2.1% | 2.6% | 2.4% |
| Trustmark B (0.95) | SSL (42dB,30bits) | - | series | 0.5 | 130 | 45.7% | 41.7 | 1.0 | 42.8% | 22.4% | 21.9% | 20.2% | 18.1% |
| Trustmark B (0.95) | SSL (42dB,30bits) | - | series | - | 130 | 92.9% | 38.4 | 1.1 | 88.7% | 54.1% | 61.1% | 53.8% | 48.6% |
| Trustmark B (0.95) | Trustmark B (0.95) | - | parallel | 0.5 | 200 | 0.0% | 42.0 | 2.1 | 0.0% | 0.0% | 0.0% | 0.0% | 0.0% |
| Trustmark B (0.95) | Trustmark B (0.95) | - | parallel | - | 200 | 0.0% | 42.0 | 2.1 | 0.0% | 0.0% | 0.0% | 0.0% | 0.0% |
| Trustmark B (0.95) | Trustmark B (0.95) | - | series | 0.5 | 200 | 0.0% | 41.4 | 2.2 | 0.0% | 0.0% | 0.0% | 0.0% | 0.0% |
| Trustmark B (0.95) | Trustmark B (0.95) | - | series | - | 200 | 0.0% | 38.0 | 2.1 | 0.0% | 0.0% | 0.0% | 0.0% | 0.0% |
| Trustmark B (0.95) | Trustmark Q (0.95) | - | series | 0.5 | 200 | 0.1% | 41.9 | 2.2 | 0.1% | 0.0% | 0.0% | 0.0% | 0.0% |
| Trustmark B (0.95) | Trustmark Q (0.95) | - | series | - | 200 | 0.2% | 38.6 | 2.3 | 0.2% | 0.0% | 0.0% | 0.0% | 0.0% |
| Trustmark Q (0.40) | - | LC[100,32,25] | - | - | 32 | 49.6% | 47.9 | 1.8 | 54.0% | 30.7% | 40.5% | 38.3% | 36.0% |
| Trustmark Q (0.45) | - | LC[100,32,25] | - | - | 32 | 65.5% | 47.3 | 1.9 | 70.5% | 41.3% | 54.9% | 51.7% | 48.5% |
| Trustmark Q (0.50) | - | LC[100,32,25] | - | - | 32 | 80.2% | 46.8 | 2.0 | 83.5% | 51.8% | 68.4% | 64.3% | 60.9% |
| Trustmark Q (0.55) | - | LC[100,32,25] | - | - | 32 | 88.9% | 46.2 | 2.0 | 91.0% | 58.7% | 77.1% | 73.8% | 70.5% |
| Trustmark Q (0.60) | - | LC[100,32,25] | - | - | 32 | 95.1% | 45.7 | 2.1 | 95.9% | 64.0% | 83.2% | 81.3% | 76.7% |

| First method | Second method | ECC | Ensemble | Clip | Capacity (bits) | Accuracy | Mean PSNR [dB] | Std PSNR [dB] | Robust. RivaGAN | Robust. SSL | Robust. TmLow | Robust. TmMed | Robust. TmHigh |
|---|---|---|---|---|---|---|---|---|---|---|---|---|---|
| Trustmark Q (0.60) | RivaGAN | LC[131,32,37] | series | -0.2 | 32 | 85.3% | 39.8 | 0.9 | 86.5% | 38.7% | 52.4% | 49.2% | 46.9% |
| Trustmark Q (0.60) | RivaGAN | LC[131,32,37] | series | -0.1 | 32 | 81.7% | 40.1 | 0.8 | 83.5% | 36.3% | 48.6% | 45.7% | 41.8% |
| Trustmark Q (0.60) | RivaGAN | LC[131,32,37] | series | 0 | 32 | 75.3% | 40.7 | 0.7 | 78.5% | 33.0% | 42.1% | 39.8% | 36.5% |
| Trustmark Q (0.60) | RivaGAN | LC[131,32,37] | series | 0.1 | 32 | 69.9% | 41.3 | 0.6 | 72.7% | 28.0% | 36.0% | 34.1% | 31.3% |
| Trustmark Q (0.60) | RivaGAN | LC[131,32,37] | series | 0.2 | 32 | 63.6% | 41.7 | 0.8 | 67.4% | 22.6% | 32.1% | 30.6% | 25.4% |
| Trustmark Q (0.60) | RivaGAN | LC[131,32,37] | series | 0.3 | 32 | 58.3% | 42.2 | 0.9 | 62.8% | 22.3% | 26.8% | 26.0% | 23.7% |
| Trustmark Q (0.60) | RivaGAN | LC[131,32,37] | series | 0.4 | 32 | 51.0% | 42.6 | 1.0 | 55.0% | 17.0% | 23.5% | 21.4% | 18.5% |
| Trustmark Q (0.60) | RivaGAN | LC[131,32,37] | series | 0.5 | 32 | 41.3% | 43.2 | 1.2 | 44.0% | 13.1% | 17.1% | 15.7% | 15.9% |
| Trustmark Q (0.60) | RivaGAN | LC[131,32,37] | series | 0.6 | 32 | 35.1% | 43.6 | 1.3 | 39.6% | 12.4% | 15.5% | 14.2% | 12.4% |
| Trustmark Q (0.60) | RivaGAN | LC[131,32,37] | series | 0.7 | 32 | 33.0% | 44.0 | 1.5 | 37.6% | 11.0% | 14.2% | 13.1% | 12.8% |
| Trustmark Q (0.60) | RivaGAN | LC[131,32,37] | series | 0.8 | 32 | 29.5% | 44.6 | 1.8 | 33.6% | 9.7% | 11.8% | 11.3% | 10.6% |
| Trustmark Q (0.60) | RivaGAN | LC[131,32,37] | series | 0.9 | 32 | 26.5% | 45.1 | 2.0 | 27.9% | 8.3% | 9.9% | 9.1% | 7.9% |
| Trustmark Q (0.60) | RivaGAN | LC[131,32,37] | series | 1 | 32 | 22.5% | 45.6 | 2.1 | 23.5% | 6.7% | 8.5% | 7.7% | 6.2% |
| Trustmark Q (0.60) | RivaGAN | LC[131,32,37] | series | 1.1 | 32 | 17.7% | 46.0 | 2.3 | 17.8% | 5.0% | 6.9% | 5.5% | 5.0% |
| Trustmark Q (0.60) | RivaGAN | LC[131,32,37] | series | 1.2 | 32 | 14.3% | 46.5 | 2.4 | 14.8% | 4.7% | 5.0% | 5.3% | 4.7% |
| Trustmark Q (0.60) | RivaGAN | LC[131,32,37] | series | - | 32 | 94.2% | 39.0 | 0.8 | 95.0% | 46.1% | 59.8% | 57.7% | 53.2% |
| Trustmark Q (0.65) | - | LC[100,32,25] | - | - | 32 | 97.8% | 45.2 | 2.1 | 98.0% | 65.1% | 87.1% | 85.0% | 82.8% |
| Trustmark Q (0.70) | - | LC[100,32,25] | - | - | 32 | 98.7% | 44.7 | 2.2 | 98.8% | 66.8% | 89.2% | 87.6% | 85.6% |
| Trustmark Q (0.75) | - | LC[100,32,25] | - | - | 32 | 99.6% | 44.2 | 2.2 | 99.5% | 68.9% | 89.8% | 88.8% | 86.7% |
| Trustmark Q (0.80) | - | LC[100,32,25] | - | - | 32 | 99.6% | 43.7 | 2.3 | 99.5% | 69.7% | 90.9% | 89.1% | 87.6% |
| Trustmark Q (0.80) | RivaGAN | LC[131,32,37] | series | -0.2 | 32 | 95.9% | 39.9 | 0.9 | 96.5% | 50.5% | 66.5% | 63.8% | 61.6% |
| Trustmark Q (0.80) | RivaGAN | LC[131,32,37] | series | -0.1 | 32 | 95.6% | 40.3 | 0.9 | 96.1% | 47.3% | 63.5% | 61.1% | 59.0% |
| Trustmark Q (0.80) | RivaGAN | LC[131,32,37] | series | 0 | 32 | 95.4% | 40.6 | 0.8 | 96.1% | 46.5% | 62.1% | 58.5% | 55.4% |
| Trustmark Q (0.80) | RivaGAN | LC[131,32,37] | series | 0.1 | 32 | 93.7% | 41.0 | 0.8 | 94.2% | 47.1% | 58.9% | 56.6% | 54.2% |
| Trustmark Q (0.80) | RivaGAN | LC[131,32,37] | series | 0.2 | 32 | 92.6% | 41.3 | 0.8 | 94.1% | 44.5% | 57.3% | 55.8% | 52.3% |
| Trustmark Q (0.80) | RivaGAN | LC[131,32,37] | series | 0.3 | 32 | 92.2% | 41.6 | 1.0 | 94.0% | 42.8% | 56.1% | 53.3% | 50.0% |
| Trustmark Q (0.80) | RivaGAN | LC[131,32,37] | series | 0.4 | 32 | 88.7% | 41.9 | 1.1 | 90.6% | 40.5% | 52.1% | 48.7% | 45.3% |
| Trustmark Q (0.80) | RivaGAN | LC[131,32,37] | series | 0.5 | 32 | 86.0% | 42.1 | 1.2 | 88.2% | 37.3% | 49.5% | 44.5% | 42.4% |
| Trustmark Q (0.80) | RivaGAN | LC[131,32,37] | series | 0.6 | 32 | 83.4% | 42.5 | 1.4 | 86.6% | 35.0% | 46.1% | 42.1% | 39.3% |
| Trustmark Q (0.80) | RivaGAN | LC[131,32,37] | series | 0.7 | 32 | 81.3% | 42.8 | 1.5 | 84.1% | 32.1% | 41.5% | 39.4% | 36.7% |
| Trustmark Q (0.80) | RivaGAN | LC[131,32,37] | series | 0.8 | 32 | 78.4% | 43.1 | 1.7 | 80.9% | 29.0% | 40.7% | 37.5% | 34.4% |
| Trustmark Q (0.80) | RivaGAN | LC[131,32,37] | series | 0.9 | 32 | 74.3% | 43.4 | 1.9 | 78.4% | 28.0% | 36.0% | 33.5% | 30.4% |
| Trustmark Q (0.80) | RivaGAN | LC[131,32,37] | series | 1 | 32 | 71.3% | 43.7 | 2.1 | 74.2% | 25.8% | 34.0% | 32.1% | 28.8% |
| Trustmark Q (0.80) | RivaGAN | LC[131,32,37] | series | 1.1 | 32 | 66.6% | 44.0 | 2.3 | 69.5% | 24.5% | 33.7% | 28.3% | 26.2% |
| Trustmark Q (0.80) | RivaGAN | LC[131,32,37] | series | 1.2 | 32 | 61.4% | 44.4 | 2.5 | 65.1% | 21.8% | 27.6% | 26.4% | 22.8% |
| Trustmark Q (0.80) | RivaGAN | LC[131,32,37] | series | - | 32 | 99.6% | 38.4 | 1.0 | 99.7% | 59.7% | 75.2% | 73.8% | 71.6% |
| Trustmark Q (0.85) | - | LC[100,32,25] | - | - | 32 | 99.8% | 43.3 | 2.3 | 99.6% | 69.8% | 91.3% | 90.3% | 89.0% |
| Trustmark Q (0.90) | - | LC[100,32,25] | - | - | 32 | 99.8% | 42.9 | 2.3 | 99.6% | 70.1% | 91.4% | 90.9% | 89.0% |
| Trustmark Q (0.95) | DwtDct | - | series | 0.5 | 132 | 15.9% | 40.3 | 2.5 | 3.9% | 1.8% | 0.0% | 0.0% | 0.0% |
| Trustmark Q (0.95) | DwtDct | - | series | - | 132 | 48.4% | 36.6 | 3.0 | 12.6% | 6.1% | 0.0% | 0.0% | 0.0% |
| Trustmark Q (0.95) | DwtDctSvd | - | series | 0.5 | 132 | 23.0% | 40.3 | 2.6 | 6.5% | 4.6% | 0.0% | 0.0% | 0.0% |
| Trustmark Q (0.95) | DwtDctSvd | - | series | - | 132 | 84.3% | 36.4 | 3.0 | 19.5% | 14.3% | 0.2% | 0.1% | 0.2% |
| Trustmark Q (0.95) | HiDDeN | - | series | 0.5 | 148 | 0.0% | 37.5 | 3.1 | 0.0% | 0.0% | 0.0% | 0.0% | 0.0% |
| Trustmark Q (0.95) | HiDDeN | - | series | - | 148 | 11.4% | 32.1 | 4.1 | 3.7% | 1.5% | 0.0% | 0.0% | 0.0% |
| Trustmark Q (0.95) | - | LC[100,32,25] | - | - | 32 | 99.9% | 42.5 | 2.4 | 99.8% | 70.3% | 92.2% | 90.9% | 89.8% |
| Trustmark Q (0.95) | - | | - | - | 100 | 93.7% | 42.5 | 2.4 | 94.8% | 62.6% | 81.0% | 77.8% | 74.3% |
| Trustmark Q (0.95) | RivaGAN | LC[131,32,37] | series | -0.2 | 32 | 99.6% | 39.9 | 1.1 | 99.5% | 56.8% | 73.8% | 70.1% | 68.2% |
| Trustmark Q (0.95) | RivaGAN | LC[131,32,37] | series | -0.1 | 32 | 99.4% | 40.1 | 1.0 | 99.2% | 53.9% | 72.7% | 69.7% | 66.5% |
| Trustmark Q (0.95) | RivaGAN | LC[131,32,37] | series | 0 | 32 | 99.1% | 40.4 | 1.0 | 99.2% | 55.2% | 71.4% | 69.9% | 66.0% |
| Trustmark Q (0.95) | RivaGAN | LC[131,32,37] | series | 0.1 | 32 | 99.2% | 40.7 | 1.0 | 98.9% | 55.2% | 69.7% | 67.4% | 64.5% |
| Trustmark Q (0.95) | RivaGAN | LC[131,32,37] | series | 0.2 | 32 | 99.2% | 40.9 | 1.0 | 98.5% | 53.2% | 67.5% | 65.7% | 62.7% |
| Trustmark Q (0.95) | RivaGAN | LC[131,32,37] | series | 0.3 | 32 | 99.2% | 41.1 | 1.1 | 99.0% | 52.2% | 67.4% | 66.8% | 62.8% |
| Trustmark Q (0.95) | RivaGAN | LC[131,32,37] | series | 0.4 | 32 | 98.7% | 41.3 | 1.2 | 98.2% | 51.0% | 66.4% | 63.9% | 61.7% |
| Trustmark Q (0.95) | RivaGAN | LC[131,32,37] | series | 0.5 | 32 | 97.9% | 41.5 | 1.2 | 97.7% | 50.2% | 64.4% | 63.0% | 55.7% |
| Trustmark Q (0.95) | RivaGAN | LC[131,32,37] | series | 0.6 | 32 | 97.4% | 41.8 | 1.3 | 97.5% | 46.7% | 61.9% | 60.6% | 56.0% |
| Trustmark Q (0.95) | RivaGAN | LC[131,32,37] | series | 0.7 | 32 | 96.7% | 42.0 | 1.5 | 96.5% | 45.8% | 61.3% | 58.1% | 53.9% |
| Trustmark Q (0.95) | RivaGAN | LC[131,32,37] | series | 0.8 | 32 | 95.6% | 42.2 | 1.6 | 95.6% | 44.8% | 57.6% | 57.6% | 52.5% |
| Trustmark Q (0.95) | RivaGAN | LC[131,32,37] | series | 0.9 | 32 | 94.4% | 42.5 | 1.8 | 94.3% | 43.1% | 56.4% | 54.3% | 48.8% |
| Trustmark Q (0.95) | RivaGAN | LC[131,32,37] | series | 1 | 32 | 92.8% | 42.7 | 1.9 | 93.3% | 40.3% | 53.4% | 51.7% | 48.9% |
| Trustmark Q (0.95) | RivaGAN | LC[131,32,37] | series | 1.1 | 32 | 91.3% | 42.9 | 2.1 | 92.0% | 41.6% | 51.6% | 50.3% | 47.4% |
| Trustmark Q (0.95) | RivaGAN | LC[131,32,37] | series | 1.2 | 32 | 89.7% | 43.2 | 2.2 | 90.0% | 36.8% | 50.3% | 47.3% | 45.1% |
| Trustmark Q (0.95) | RivaGAN | LC[131,32,37] | series | - | 32 | 99.9% | 38.0 | 1.1 | 100.0% | 59.6% | 81.3% | 77.5% | 75.3% |
| Trustmark Q (0.95) | RivaGAN | | series | 0.5 | 132 | 17.3% | 41.5 | 1.2 | 20.0% | 3.8% | 6.1% | 5.1% | 4.4% |
| Trustmark Q (0.95) | RivaGAN | | series | - | 132 | 67.3% | 38.0 | 1.1 | 66.1% | 16.8% | 24.6% | 22.5% | 18.8% |
| Trustmark Q (0.95) | RoSteALS | | series | 0.5 | 200 | 1.1% | 36.8 | 2.8 | 0.3% | 0.0% | 0.0% | 0.0% | 0.0% |
| Trustmark Q (0.95) | RoSteALS | | series | - | 200 | 14.5% | 30.8 | 3.4 | 7.5% | 4.4% | 0.0% | 0.0% | 0.0% |
| Trustmark Q (0.95) | SSL (42dB,30bits) | | series | 0.5 | 130 | 35.5% | 42.3 | 1.2 | 33.4% | 15.9% | 14.7% | 14.5% | 12.6% |
| Trustmark Q (0.95) | SSL (42dB,30bits) | | series | - | 130 | 85.4% | 39.0 | 1.1 | 83.3% | 45.8% | 54.8% | 49.5% | 44.5% |
| Trustmark Q (0.95) | Trustmark B (0.95) | | parallel | 0.5 | 200 | 0.0% | 43.1 | 2.1 | 0.0% | 0.0% | 0.0% | 0.0% | 0.0% |
| Trustmark Q (0.95) | Trustmark B (0.95) | | parallel | - | 200 | 0.0% | 43.1 | 2.1 | 0.0% | 0.0% | 0.0% | 0.0% | 0.0% |
| Trustmark Q (0.95) | Trustmark B (0.95) | | series | 0.5 | 200 | 0.0% | 41.9 | 2.2 | 0.0% | 0.0% | 0.0% | 0.0% | 0.0% |
| Trustmark Q (0.95) | Trustmark B (0.95) | | series | - | 200 | 0.0% | 38.7 | 2.2 | 0.0% | 0.0% | 0.0% | 0.0% | 0.0% |
| Trustmark Q (0.95) | Trustmark Q (0.95) | | parallel | 0.5 | 200 | 0.0% | 42.9 | 2.3 | 0.0% | 0.0% | 0.0% | 0.0% | 0.0% |
| Trustmark Q (0.95) | Trustmark Q (0.95) | | parallel | - | 200 | 0.0% | 42.9 | 2.3 | 0.0% | 0.0% | 0.0% | 0.0% | 0.0% |
| Trustmark Q (0.95) | Trustmark Q (0.95) | | series | 0.5 | 200 | 0.0% | 42.5 | 2.3 | 0.0% | 0.0% | 0.0% | 0.0% | 0.0% |
| Trustmark Q (0.95) | Trustmark Q (0.95) | | series | - | 200 | 0.0% | 39.0 | 2.4 | 0.0% | 0.0% | 0.0% | 0.0% | 0.0% |
| Trustmark Q (1.00) | - | LC[100,32,25] | - | - | 32 | 99.9% | 42.1 | 2.4 | 99.8% | 71.3% | 92.0% | 91.1% | 90.1% |

