# OpenReview forum: "On the Coexistence and Ensembling of Watermarks"
_ICLR.cc/2025/Conference — Submitted to ICLR 2025_

### Official Review · Reviewer_HNTi · 2024-10-31

**Soundness:** 3
**Presentation:** 3
**Contribution:** 3
**Rating:** 6
**Confidence:** 5

**Summary:**

This manuscript starts from the consideration of two possible scenarios that need to add several watermarks from different sources into the same carrier: the addition of  super watermark to inform the user which watermark detector to use for this image, and different users would use different watermarks for their content provenance goals. Based on these demands, authors point out the need and importance for different watermarks to simultaneously exist in the same carrier. Thus, authors conduct ample experiments on various watermarking methods to add the new watermark information into the image containing another watermark message to check whether or not watermark messages from various sources can co-exist in the same image.  The experimental result shows that, different from the expectation of many peoples that watermarks may overwrite each other, in most situations, after adding the second watermark, the first watermark message can still be decoded with a high accuracy even if confinements are made to avoid the degradation of visual quality or robustness caused by the existence of two watermarks. And authors claim that to their knowledge, watermark coexistence has not been studied before. With this finding, the manuscript proposes to utilize the watermark coexistence to modify the performance of existing methods without needing any retraining, and sometimes this may bring the enhancement of some factors in capacity, accuracy, robustness, and visual quality.

**Strengths:**

1. As the claim of authors, this manuscript firstly try to explore the watermark coexistence, which is a valuable attempt and interesting perspective in the field of watermarking, and this work may inspire more researchers to involve in the filling of existing research blanks of watermarking technology.
2. It is good for authors of this manuscript to find out two persuasive scenarios to demonstrate the demand to contain multiple watermarks in a same coverage, which can contribute to the application meaning of the research in this manuscript, and demonstrate authors' good understanding about the application situation of watermarking technology in the real scenario.
3. To verify that different watermarks can co-exist simultaneously, authors conducted an ample amount of experiments to provide sufficient experimental evidence. The considerable volume of work completed by authors is an advantage of this work.
4. After finding the watermark coexistence, authors propose two straightforward methods to add multiple watermarks into the image: serial method and parallel method. Even if they are easy methods without the complex design, they can actually be implemented by users with the need to embed several different watermarks, so as to directly meet the related demand, obtaining certain application value.
5. The organization of this paper is clear and readable. Authors starts from the demand in the real scenario, then use experiments to verify the existence of watermark coexistence, and the analysis about the experimental can also help to illustrate the research of this manuscript. After that, methods to utilize the watermark coexistence are proposed, with experiments conducted to test the performance. The thinking of authors when writing this manuscript can be easily captured, which is beneficial for readers to better understand this work.

**Weaknesses:**

1. In the Introduction of this manuscript, authors claim that the same principles found in this manuscript likely apply also to video, audio, and to some extent, text watermarking. However, this paper just focus on the image watermarking methods, which fails to give persuasive evidence to illustrate this point. So authors can modify the expression of this manuscript to just focus on image watermarking, or consider to add some experiments using audio or text watermarking methods to prove their opinion.

2. As written by authors, this manuscript can successfully find out the watermark coexistence phenomenon, but fails to figure out the root causes of this phenomenon. Authors are encouraged to give some theoretical analysis to get closer to the real factor causing the watermark coexistence. This is also beneficial for the improvement of academic meaning of this manuscript.

3. This manuscript's findings and contributions are significantly based on the experiments or the reutilization about existing methods. Even if this still has its meaning, the further exhibition about the novelty of this manuscript  is inevitably affected. It would be better for authors to go further with their acquired findings shown in this manuscript and better strengthen the novelty of their work.

4. In the experiment, authors find that in some cases, a strong base method with ECC can perform better than ensembling. This will do harm to the convenient utilization of ordinary users about adding different watermarks, because it will not be so easy for ordinary users to decide whether or not the combination of watermarking methods they want to use can definitely achieve ideal performance they desire compared to returning to a strong base method with ECC.

5. The spelling mistake should  be eliminated in the next version of this manuscript. For example, in about line 63, the shown "neeeding" should be corrected to "needing".

**Questions:**

1. There lacks the evidence to prove the watermark coexistence can still be seen in other categories of carriers different from images. Authors can modify the expression of this manuscript to just focus on image watermarking, or consider to add some experiments using audio or text watermarking methods to prove their opinion.

2. Authors are encouraged to give more theoretical analysis to try to find out the root reason causing the watermark coexistence, which is also beneficial for the improvement of academic value of this manuscript.

3. Based on their acquired findings shown in this manuscript, it would be better for authors to go further and show more special contributions less relying on previous achievements, so as to  strengthen the novelty of their work.

4. Authors should manage to find a method to better instruct ordinary users to correctly find watermarking methods that can really achieve desired performance with the utilization of watermark coexistence, compared to just use one strong base method with ECC.

5. Typos should  be fixed after the revision of this manuscript. For example, in about line 63, the shown "neeeding" should be changed to "needing".

---

> ### Author Response · Authors · 2024-11-21
>
> We thank the reviewer for their detailed and constructive feedback. We appreciate their recognition of the novelty of our work in exploring watermark coexistence, the real-world relevance of our proposed scenarios, and the clarity of our paper's organisation. With regards to their concerns and questions, here are our answers:
>
> > __W1 / Q1: There lacks the evidence to prove the watermark coexistence can still be seen in other categories of carriers different from images. Authors can modify the expression of this manuscript to just focus on image watermarking, or consider to add some experiments using audio or text watermarking methods to prove their opinion.__
>
> As we disclose at the end of the Introduction (lines 65–66), we restrict the focus of the present work to watermarking in the image domain. However, we agree with the reviewer that some parts of the paper might sound more general than we aimed them to be therefore. Therefore, we have edited the Abstract and the Introduction to be more explicitly clear that we are focusing on image watermarking. Thank you for your feedback!
>
> > __W2 / Q2. Authors are encouraged to give more theoretical analysis to try to find out the root reason causing the watermark coexistence, which is also beneficial for the improvement of academic value of this manuscript.__
>
> We thank the reviewer for their suggestion. We have added a new Appendix A providing a geometric intuition behind watermark coexistence.  While a rigorous theoretical analysis is beyond the scope of this work, our geometric interpretation offers insights into the underlying mechanisms that enable different watermarking methods to coexist. We believe this analysis enhances the academic value of our manuscript.
>
> > __W3 / Q3. Based on their acquired findings shown in this manuscript, it would be better for authors to go further and show more special contributions less relying on previous achievements, so as to strengthen the novelty of their work.__
>
> The novelty of our work is not in proposing a single novel watermarking model but rather in showing that there can exist watermarking systems consisting of multiple watermarking methods. The core contributions of our work lie in demonstrating the feasibility of watermark coexistence for modern deep learning-based watermarking techniques. We are the first to formally introduce this concept, evaluate its effectiveness, and propose ensembling as a novel method to enhance watermark capacity. Our work provides a deeper understanding of the underlying mechanisms of watermark coexistence and offers practical strategies for leveraging this phenomenon to achieve improved performance trade-offs.
>
> > __W4 / Q4. Authors should manage to find a method to better instruct ordinary users to correctly find watermarking methods that can really achieve desired performance with the utilization of watermark coexistence, compared to just use one strong base method with ECC.__
>
> We acknowledge the reviewer's concern regarding the potential complexity of choosing optimal watermarking methods for ordinary users. However, it's important to note that in practical scenarios, end-users often don't directly select watermarking methods. Instead, they rely on systems or platforms that incorporate watermarking as a backend process. Our work is primarily targeted at the developers of these systems. By understanding the trade-offs between different methods and their potential for coexistence, these developers can make informed decisions to optimize watermarking performance for their specific use cases.
>
> > __W5 / Q5. Typo__
>
> Thank you for spotting this typo. We have fixed it.

---

> > ### Comment · Reviewer_HNTi · 2024-11-27
> > **Reviewer's comments**
> >
> > Thanks for your feedback.

---

### Official Review · Reviewer_aCCB · 2024-11-01

**Soundness:** 3
**Presentation:** 3
**Contribution:** 3
**Rating:** 6
**Confidence:** 4

**Summary:**

This paper analyzes the coexistence of deep-learning-based watermarking schemes and proposes leveraging this coexistence to enable ensemble watermarking models. The authors emphasize the significance of supporting multiple watermarking schemes in practical applications and explore this through extensive experimentation. However, based on the reported results, the benefits of ensembling do not appear clearly advantageous when considering the capacity-robustness-quality trade-off.

**Strengths:**

1. This paper investigates the coexistence of deep-learning-based watermarking schemes. The coexistence problem is an important aspect for practical application.
2. The study provides comprehensive experimental analyses on the coexistence and ensembling of watermarks. These results are valuable for further studies.

**Weaknesses:**

1. The conditions for robustness testing are not fully explained. Section 2 contains ambiguous phrases, such as "augmentations used for training RivaGAN." Clear details on the types of attacks used, each attack's strength, and the methodology for calculating the robustness values would improve clarity.
2. The effectiveness of ensembling under varying attack strengths needs further verification. Given that robustness is achieved through ECC, it would be beneficial to assess whether the proposed method can withstand strong attacks.
4. Despite the inclusion of ECC, the observed robustness values appear insufficient for practical applications.
3. Figure 5 does not demonstrate an improvement in the overall capacity-robustness-quality trade-off with ensembling. For example, in sub figure B, ensembling reduces robustness compared to TrustMark. Although a slight improvement in quality is noted, it does not suggest that ensembling offers a better trade-off.

**Questions:**

Please refer to the Weakness.

---

> ### Author Response · Authors · 2024-11-21
>
> We thank the reviewer for acknowledging the practical importance of the watermark coexistence problem and the value to the field of our comprehensive experimental analysis. Here are our responses to the reviewer’s concerns:
>
> > __W1. The conditions for robustness testing are not fully explained. Section 2 contains ambiguous phrases, such as "augmentations used for training RivaGAN." Clear details on the types of attacks used, each attack's strength, and the methodology for calculating the robustness values would improve clarity.__
>
> The reviewer is right, the nature of the augmentations for the various robustness metrics is an important detail and we apologise for the omission. We have added these details to a new Appendix D in the revised manuscript. Thank you for pointing this out!
>
> > __W2. The effectiveness of ensembling under varying attack strengths needs further verification. Given that robustness is achieved through ECC, it would be beneficial to assess whether the proposed method can withstand strong attacks.__
>
> We would like to clarify that robustness is not achieved solely through error-correcting codes (ECCs) but also via the inherent robustness of the base watermarking models. ECCs are only used to further boost the robustness. We are not sure what the reviewer considers “strong attacks” but we evaluate against 5 different sets of augmentations and observe that for every model there is at least one set of augmentations that the model would have less than 50% accuracy for. Hence, the augmentations we consider appear to be sufficiently strong and challenging for the 8 models we consider.
>
> > __W3. Despite the inclusion of ECC, the observed robustness values appear insufficient for practical applications.__
>
> We are not sure what level of robustness the reviewer considers sufficient for practical applications, but in our experience that depends greatly on the application and what trade-off it can tolerate. That is a key thesis of our paper, and a key property of watermarking: it is all about the trade-offs. Using the techniques in the paper, we can easily construct very robust watermarks, by reducing the image quality or the capacity (e.g. we can apply the trivial repetition code to a 100bit block size to get 1bit message, while being able to correct up to 49 bit flips). We chose the case studies in the paper with a somewhat balanced trade-off setting in mind, but of course, any other setting is also possible.
>
> > __W4. Figure 5 does not demonstrate an improvement in the overall capacity-robustness-quality trade-off with ensembling. For example, in sub figure B, ensembling reduces robustness compared to TrustMark. Although a slight improvement in quality is noted, it does not suggest that ensembling offers a better trade-off.__
>
> We want to clarify that we do not claim that we have experimental results showing that ensembling _simultaneously_ improves capacity, accuracy, robustness and quality. In fact, we state the opposite (see lines 508-511). What we claim is that ensembling can enable _different_ trade-offs than what the original watermarking model can do alone. In this way, ensembling is a powerful tool for specialising existing methods for the concrete requirements of downstream applications.

---

> > ### Comment · Reviewer_EYaB · 2024-11-24
> >
> > Thanks for your responses, all of my concerns have been addressed. I have adjusted the score.

---

> > > ### Author Response · Authors · 2024-11-26
> > >
> > > Thank you so much! We especially appreciate your feedback on the explanations and the need for better theoretical grounding. We believe they helped improve our paper significantly.

---

> > ### Comment · Reviewer_aCCB · 2024-11-25
> >
> > Thank the authors for the detailed response. I agree that the authors do not claim in their paper that ensembling enables a better trade-off, and I appreciate your openness in discussing this point. However, I frequently raise concerns about robustness because the degradation introduced by ensembling (as shown in Figures 6 and 8) is an important limitation that cannot be overlooked. While ensembling offers more trade-off options, the degraded quality of these trade-offs may restrict its practical applicability. Overall, I like this paper. And taking everything into account, I maintain my original borderline accept score.

---

> > > ### Author Response · Authors · 2024-11-26
> > >
> > > Thank you for the careful reading of our work and for sharing your feedback! Although we would love for that to not be the case, we agree that robustness is indeed hurt by ensembling, sometimes irrecoverably so. It is the cost we need to pay for co-existence. We hope that follow-up works would try to improve on that, possibly by further fine-tuning (as suggested in this discussion), via more advanced error-correcting, or by specifically training models to produce watermarks that coexist well. At any rate, we understand why you wish to keep your score as is.

---

### Official Review · Reviewer_EYaB · 2024-11-01

**Soundness:** 3
**Presentation:** 3
**Contribution:** 2
**Rating:** 8
**Confidence:** 4

**Summary:**

This paper studies the coexistence problem of image watermarking methods and explores how to improve the overall information capacity and achieve new trade-offs between quality, capacity, accuracy and robustness by ensembling watermarking methods. Research shows that: 1. although multiple watermarks applied to the same image will lead to a slight decrease in image quality and decoding robustness, different watermarking methods can coexist in the same image and have little impact on their respective decoding accuracy. And 2. by ensembling watermarking methods, the message capacity of the image may increased without retraining the model.

**Strengths:**

The authors propose and verify the hypothesis that multiple deep learning-based watermarking methods can coexist, providing technical support for the practical need of multi-party watermarking usage. Additionally, through watermark ensembling, the authors demonstrate how capacity can be increased without retraining the base models, offering flexibility for the practical application of watermarking methods. The paper is logically organized and clearly articulated, addressing relevant real-world needs. The experimental design is comprehensive, exploring various combinations of watermarking methods and their impact on image quality, decoding accuracy, and robustness, with persuasive results.

**Weaknesses:**

1.The coexistence of multiple watermarks was proposed and discussed in the past as an early issue. As the author mentioned, the frequency-based method (one category of watermark coexistence research) and similar studies like [1] . Therefore, when stating the contribution, it would be better to specifically refer to deep learning-based watermark coexistence research.
2. Although the author claims that past frequency-based watermark coexistence research is not applicable to current deep learning-based watermark methods, there is a lack of corresponding discussion and formula derivation. It might be beneficial to include a comparison of relevant methods in the experiments to further enhance persuasiveness.
3. The key figures and table in the paper are very complex. Consider adding more detailed explanations to help readers understand them better.

[1]Xu T, Shao X, Yang Z. Multi-watermarking scheme for copyright protection and content authentication of digital audio[C]//Advances in Multimedia Information Processing-PCM 2009: 10th Pacific Rim Conference on Multimedia, Bangkok, Thailand, December 15-18, 2009 Proceedings 10. Springer Berlin Heidelberg, 2009: 1281-1286.

**Questions:**

As described in the weakness.

---

> ### Author Response · Authors · 2024-11-21
>
> We thank the reviewer for their kind words regarding the practical value, logical organisation, clear articulation, and comprehensive experimental design of our work. As for their outstanding concern, we have the following response:
>
> > __W1. The coexistence of multiple watermarks was proposed and discussed in the past as an early issue. As the author mentioned, the frequency-based method (one category of watermark coexistence research) and similar studies like [1] . Therefore, when stating the contribution, it would be better to specifically refer to deep learning-based watermark coexistence research.__
>
> While we tried to highlight previous work on non-deep learning-based coexistence and ensembling, that was only in the coexistence section (Section 3, lines 143–145). We agree that we should highlight that our paper focuses on deep-learning based techniques and hence have edited the abstract and introduction to highlight that. Thank you for this feedback!
>
> > __W2. Although the author claims that past frequency-based watermark coexistence research is not applicable to current deep learning-based watermark methods, there is a lack of corresponding discussion and formula derivation. It might be beneficial to include a comparison of relevant methods in the experiments to further enhance persuasiveness.__
>
> Unfortunately, the frequency-based watermark coexistence work that we have seen, including the work referenced by the reviewer, appears to be tailored for a specific base watermarking technique, e.g., DWT. Therefore, it is difficult to compare against these prior works, considering we cannot apply their techniques to most modern deep watermarking methods of interest. We are also afraid that deep-learning methods are not quite amenable to closed-form analysis and we are not sure how one could go about deriving formulas for their performance. Nevertheless, if the reviewer has concrete suggestions how we could go about this, we would appreciate their insight!
>
> Following the reviewers’ questions, we also added a new Appendix A providing a geometric perspective that helps explain why watermarking methods can coexist. While this is not a formal analysis, we hope it adds more intuition and depth to our results.
>
>
> > __W3. The key figures and table in the paper are very complex. Consider adding more detailed explanations to help readers understand them better.__
>
> Thank you for the feedback! As we aim for clear and self-contained figures, we edited the descriptions of Table 1 and Figures 2 and 3 to add further details. We hope that improves the readability of the paper.

---

### Official Review · Reviewer_wgN1 · 2024-11-04

**Soundness:** 3
**Presentation:** 3
**Contribution:** 3
**Rating:** 6
**Confidence:** 5

**Summary:**

This paper provides a pioneering investigation into the coexistence and ensembling of watermarking methods, demonstrating that multiple watermarks can coexist in a single image with minimal impact on image quality and decoding accuracy.

**Strengths:**

1. The paper offers a unique contribution by exploring the coexistence of multiple watermarks within the same image, a topic not previously studied, challenging the assumption that overlapping watermarks would overwrite each other.

2. The paper conducts extensive experiments to validate the coexistence of different watermarks and proposes an effective method for watermark ensembling using strength control and error-correcting codes.

3. The writing of this paper is generally good and easy to understand.

**Weaknesses:**

1. The authors in this work did not present complete scientific innovation; instead, it focuses more on engineering improvements and exploration, as strength clipping and error correction are relatively mature techniques. I am somewhat concerned about whether the authors have demonstrated sufficient scientific innovation in this work.

2. The authors are encouraged to provide some theoretical justification as to why most watermarking methods can coexist. Additionally, they could offer paradigms, such as which types of watermarking integrations tend to yield better results. For example, it would be helpful to investigate whether watermarks using different hiding techniques, such as frequency and spatial domains, are more easily integrated. This would aid in the design of future watermarking methods.

3. For the initially embedded watermark, the second watermark added actually serves as a form of degradation. Have the authors considered jointly training the encoding and decoding processes of the first and second watermarks? This might lead to better coexistence results.

**Questions:**

Please refer to the weakness.

---

> ### Author Response · Authors · 2024-11-21
>
> We would like to thank the reviewer for appreciating how our work leverages extensive experimentation to challenge common beliefs about the coexistence of watermarks, as well as the presentation of our paper. With that in mind, we would like to address their concerns.
>
> >__W1. The authors in this work did not present complete scientific innovation; instead, it focuses more on engineering improvements and exploration, as strength clipping and error correction are relatively mature techniques. I am somewhat concerned about whether the authors have demonstrated sufficient scientific innovation in this work.__
>
> We agree with the reviewer that strength clipping and error-correction are not novel techniques in the context of watermarking, as described in the introduction of Section 4 (line 351) and in Figure 4. The scientific contribution of our work lies elsewhere, namely:
>
> 1. We are first to formulate the need for watermark coexistence, first to evaluate it for modern deep-learning based watermarking techniques, and first to demonstrate that watermark coexistence does indeed happen out of the box.
>
> 2. We study the “fine print” of these findings, including when coexistence happens, and what trade-offs are involved, and offer potential explanations as to why it might be happening.
>
> 3. We offer ensembling watermarks, a technique which, to the best of our knowledge, has never before been applied to watermarking. We show that naïvely applying ensembling is not enough and demonstrate the importance of combining it with strength clipping and error-correction. These two are tools we use but coexistence and ensembling are the core scientific innovations of our work.
>
> Overall, the core scientific innovation of our work is showing that, contrary to popular belief, watermarks can coexist, as well as showing how this coexistence can be leveraged for obtaining new trade-offs between capacity, accuracy, robustness and image quality.
>
>
> > __W2. The authors are encouraged to provide some theoretical justification as to why most watermarking methods can coexist. Additionally, they could offer paradigms, such as which types of watermarking integrations tend to yield better results. For example, it would be helpful to investigate whether watermarks using different hiding techniques, such as frequency and spatial domains, are more easily integrated. This would aid in the design of future watermarking methods.__
>
> We thank the reviewer for their suggestion. We have added a new appendix (App. A in the revised manuscript) discussing our geometric intuition behind watermark coexistence. To quote from there:
>
> > One explanation is that watermarks are designed to be robust to various augmentations, which induces sparsity into what perturbations can be considered valid watermarks. However, there are many possible sparse patterns that can satisfy the robustness requirements and different watermarking methods might be —either by random selection, or due to particular design decisions— picking different such patterns. As a result, the perturbations used by different methods could be non-overlapping and might be sufficiently different so that they are still identifiable even after composing.
>
> While a rigorous theoretical analysis remains an open challenge, our geometric interpretation provides a conceptual understanding of why different watermarking methods can coexist. We believe this analysis supports our empirical findings and provides a foundation for future research in this area.
>
>
> > __W3. For the initially embedded watermark, the second watermark added actually serves as a form of degradation. Have the authors considered jointly training the encoding and decoding processes of the first and second watermarks? This might lead to better coexistence results.__
>
> This is an excellent observation! We did consider fine-tuning the individual watermarking models to work better when applied together. We also mention that in the Discussion section of the paper (line 533). However, we ultimately decided that this would be out-of-scope for the current paper. First, if we modify the models, analysing whether the models coexist would be much more complicated as there would be a dependency on a training dataset and hyperparameters. It would also not help us with making conclusions about the original models. Furthermore, some of the methods we considered cannot be fine-tuned without drastically modifying the method itself. For example, DwtDct and DwtDctSvd have no learnable parameters, while SSL is not trained as it utilises a frozen embedding model. Hence, fine-tuning these models might change them too much. Thus, we concluded that this would be of little scientific value. That being said, for practical applications, we do believe there could be added utility of jointly fine-tuning the constituent methods, if that is possible.

---

> ### Comment · Reviewer_wgN1 · 2024-11-27
>
> Thank you for your responses. I have carefully reviewed all the comments and your replies. My main concerns have been addressed, so I will maintain my original score. Good luck!

---

### Comment · Area_Chair_J3xB · 2024-11-23

Dear Reviewers,
The authors have responded to your valuable comments.
Please take a look at them!

Best,
AC

---

### Meta-Review · Area_Chair_J3xB · 2024-12-18

**Metareview:**

As claimed by the authors, they ``first studied the coexistence of deep image watermarking methods and, contrary to intuition, they find that various open-source watermarks can coexist with only minor impacts on image quality and decoding robustness.'' and ``showed how ensembling can increase the overall message capacity and enable new trade-offs between capacity, accuracy, robustness and image quality, without needing to retrain the base models.''
The main contribution is to explore the coexistence of multiple watermarks within the same image.

However, the weaknesses include: (1) what kind of applications that needs to increase message capacity is not clearly specified in the paper. For example, for copyright protection purpose, the message/watermark capacity is not relevant to robustness. (2) lack of theoretical support (Reviewer HNTi). This study basically is an experiment work. (3) lack of robustness evaluation. The authors did not clearly describe how the used watermarking method can resist geometric distortions, as described in Appendix D. In particular, for example, for TrustMark, there were no geometric attacks, such as rotations, scaling, cropping, and any combination of them, used in the experiments.

Moreover, in responding to W2 of Reviewer aCCB, the authors replied that ``We would like to clarify that robustness is not achieved solely through error-correcting codes (ECCs) but also via the inherent robustness of the base watermarking models. ECCs are only used to further boost the robustness.''
But in responding to W3 of Reviewer aCCB, the authors replied ``...we can easily construct very robust watermarks, by reducing the image quality or the capacity (e.g. we can apply the trivial repetition code to a 100bit block size to get 1bit message, while being able to correct up to 49 bit flips).'' => the use of ECC.

Overall, the critical weakness of this submission is to overlook robustness. And its evaluation on geometric (stronger) attacks is questionable!

**Additional Comments On Reviewer Discussion:**

During the rebuttal period, the authors have properly responded to reviewers' comments.
However, for Reviewer aCCB's comment on ``...However, I frequently raise concerns about robustness because the degradation introduced by ensembling (as shown in Figures 6 and 8) is an important limitation that cannot be overlooked...,'' the authors still did not clearly reply how they deal with robustness, in particular, for resisting geometric distortions.
Robustness is a very important requirement when the tradeoff among fidelity, capacity, and robustness is concerned.

---

### Decision · Program_Chairs · 2025-01-22

Reject